JOURNAL OF
Neuroscience Research

# Extracellular microvesicles promote microglia-mediated pro-inflammatory responses to ethanol

Fulton T. Crews[1,2,3]    |    Jian Zou[1]    |    Leon G. Coleman Jr.[1,2] (ID)

[1]Bowles Center for Alcohol Studies, The University of North Carolina at Chapel Hill, School of Medicine, Chapel Hill, NC, USA

[2]Department of Pharmacology, The University of North Carolina at Chapel Hill, School of Medicine, Chapel Hill, NC, USA

[3]Department of Psychiatry, The University of North Carolina, School of Medicine, Chapel Hill, NC, USA

**Correspondence**
Leon G. Coleman, Jr., Bowles Center for Alcohol Studies, The University of North Carolina at Chapel Hill, The School of Medicine, CB #7178, 1007C Thurston-Bowles Building, Chapel Hill, NC 27599-7178, USA.
Email: leon_coleman@med.unc.edu

**Funding information**
National Institutes of Health, National Institute on Alcohol Abuse and Alcoholism, Grant/Award Number: P60AA011605, U01AA020023, U24AA020024, T32AA007573, K08AA024829 and K08AA024829S1; Bowles Center for Alcohol Studies

## Abstract

Alcohol use disorder (AUD) pathology features pro-inflammatory gene induction and microglial activation. The underlying cellular processes that promote this activation remain unclear. Previously considered cellular debris, extracellular vesicles (EVs) have emerged as mediators of inflammatory signaling in several disease states. We investigated the role of microvesicles (MVs, 50 nm–100 μm diameter EVs) in pro-inflammatory and microglial functional gene expression using primary organotypic brain slice culture (OBSC). Ethanol caused a unique immune gene signature that featured: temporal induction of pro-inflammatory TNF-α and IL-1β, reduction of homeostatic microglia state gene Tmem119, progressive increases in purinergic receptor P2RY12 and the microglial inhibitory fractalkine receptor CX3CR1, an increase in the microglial presynaptic gene C1q, and a reduction in the phagocytic gene TREM2. MV signaling was implicated in this response as reduction of MV secretion by imipramine blocked pro-inflammatory TNF-α and IL-1β induction by ethanol, and ethanol-conditioned MVs (EtOH-MVs) reproduced the ethanol-associated immune gene signature in naïve OBSC slices. Depletion of microglia prior to ethanol treatment prevented pro-inflammatory activity of EtOH-MVs, as did incubation of EtOH-MVs with the HMGB1 inhibitor glycyrrhizin. Ethanol caused HMGB1 secretion from cultured BV2 microglia in MVs through activation of PI3 kinase. In summary, these studies find MVs modulate pro-inflammatory gene induction and microglial activation changes associated with ethanol. Thus, MVs may represent a novel therapeutic target to reduce neuroinflammation in the setting of alcohol abuse or other diseases that feature a neuroimmune component. [Correction added on 5 April 2021, after first online publication: The copyright line was changed.]

**KEYWORDS**
alcohol use disorder, extracellular vesicles, inflammation, microglia, neuroimmune

Edited by Alex S. Marshall and Cristina Ghiani. Reviewed by Gregg Homanics and Terrence Deak.

# 1 | INTRODUCTION

Alcohol use disorder (AUD) pathology involves progressive and increasing induction of inflammatory signaling in brain (Crews et al., 2017; Mayfield et al., 2013; Montesinos et al., 2016). Multiple studies have identified a role of immune signaling in AUD pathology. Transcriptomic studies in both human postmortem brain and chronic ethanol treatments in mice find induction and dysregulation of pro-inflammatory signaling pathways (Brenner et al., 2019; McCarthy et al., 2018). This pro-inflammatory activation might contribute to cellular and synaptic pathology as well as behavioral phenotypes associated with AUD (Warden et al., 2020). Pro-inflammatory immune activation in brain involves pro-inflammatory polarization of microglia (He & Crews, 2008; McCarthy et al., 2018), Toll-like receptor (TLR) signaling (Alfonso-Loeches et al., 2010), and release of TLR-activating danger-associated molecular pattern molecules (DAMPs). Studies using TLR knockout mice and TLR agonists find that TLRs are involved in both gross AUD neuropathology and alcohol self-administration (Grantham et al., 2020; McCarthy et al., 2017; Pascual et al., 2011). Recently, microglia have been found to play a key role in alcohol self-administration and synaptic changes in cortex and amygdala (Warden et al., 2020). Studies in our group suggest that release of TLR-activating DAMPs contributes to alcohol-mediated induction of inflammatory signaling (Coleman et al., 2017; Crews et al., 2013). Several cellular processes are altered by ethanol and may interact with pro-inflammatory activation. These cellular changes include alterations in features such as membrane fluidity and composition (Aloia et al., 1985; Goldstein, 1986; Harris & Hitzemann, 1981) and mitochondrial metabolism and generation of reactive oxygen species (Collins & Neafsey, 2012; Qin & Crews, 2012). Changes in several cellular processes occur with neuroimmune activation; however, the underlying drivers of pro-inflammatory activation due to ethanol remain elusive, as ethanol has no single "receptor." Identifying such a mechanism could provide new therapeutic approaches that target this long-term enhancement of inflammatory activation.

The signaling pathways associated with ethanol-induced neuroinflammation involve complex cascades resulting in coordinated regulation of several genetic programs. Initiation of this signaling involves secretion of TLR-activating DAMPs such as HMGB1 and miRNA let-7b (Coleman, Zou, & Crews, 2017; Crews et al., 2013; Zou & Crews, 2014). TLR ligation leads to activation of pro-inflammatory gene regulating transcription factors such as nuclear factor kappa-light-chain-enhancer of activated B cells (NF-κB), interferon regulatory factor 7, and AP-1 (Cui et al., 2014) (Crews, Lawrimore, et al., 2017). These transcription factors lead to increased gene expression of several pro-inflammatory cytokines such as TNF-α and IL-1β. TLR activation also causes structural as well as functional changes in microglia, the predominant immune cell in the brain (Fernandez-Lizarbe et al., 2009; Lawrimore & Crews, 2017; Rosenberger et al., 2014). At rest, microglia adopt a homeostatic/surveillance state that can change to hyper-ramified or amoeboid structural states that may feature a range of functional changes. Expression levels of microglia-enriched factors such as Tmem119, P2RY12, CX3CR1, C1q, C3R, and TREM2 (Butovsky et al., 2014) can change with different insults, exposures, or disease states and modulate different microglial functions. For instance, expression of Tmem119 and P2RY12 decline

### Significance

This work finds that microvesicles (MVs), unique small subcellular components, contribute to pro-inflammatory activation in brain by ethanol. Pro-inflammatory signaling in brain is a key component of neuropathology in alcohol use disorder. We find ethanol causes the release of inflammation-promoting MVs, and that blockade of MV release prevents pro-inflammatory signaling by ethanol. Alcohol-induced inflammation in brain shares many common features with other chronic brain conditions such as Alzheimer's disease and aging. Therefore, these mediators could represent a novel therapeutic target to prevent or reduce inflammation due to alcohol or in other neurological disease settings.

in mouse models of Alzheimer's disease (Keren-Shaul et al., 2017), C1q is involved in microglial pruning of synapses and microglial activation of astrocytes (Hong et al., 2016; Stephan et al., 2013), CX3CR1 regulates microglia activation state via interaction with neuron-derived fractalkine (CX3CL1) (Paolicelli et al., 2014), and TREM2 is induced in Alzheimer's models and is involved in microglia phagocytosis (Keren-Shaul et al., 2017). Ethanol is known to cause temporal and dynamic changes in microglia structure and activation, though its effects on these key mediators are unknown.

Extracellular vesicles (EVs) are emerging as fundamental drivers of pro-inflammatory signaling across organ systems and diseases (Buzas et al., 2014; Raeven et al., 2018; Tkach & Thery, 2016). EVs are small, membrane-enclosed structures capable of transmitting intercellular signals through the delivery of bioactive cell-derived lipids, proteins, and nucleic acids either on their surface or enclosed within their membrane. The three major classes of EVs are exosomes (50–150 nm diameter), microvesicles (MVs, 50 nm–1 μm), and apoptotic bodies (up to 5 μm) (van Niel et al., 2018). Exosomes and MVs differ in their biogenesis, though there is overlap in their sizes from ~50 to 150 nm and surface markers. Exosomes originate from the release of endosome-containing multi-vesicular bodies, while MVs primarily originate from budding from the cell surface. MVs and exosomes can carry miRNAs, mRNAs, and protein cargo. Ethanol can alter contents of both exosomes (Ibanez et al., 2019) and MVs (Coleman, Zou, & Crews, 2017, 2020). Recent evidence supports a role for EV signaling in glial to neuronal interactions during pro-inflammatory activation in response to ethanol (Coleman, Zou, & Crews, 2017; Ibanez et al., 2019; Pascual et al., 2020). However, a role for MVs in governing glial activation by ethanol is unknown.

We previously reported that MVs house DAMPs such as HMGB1 and let-7b which are TLR agonists that could activate glia (Coleman Jr. et al., 2018; Coleman, Zou, & Crews, 2017). Therefore, we hypothesized that MVs are fundamental mediators of pro-inflammatory activation in response to ethanol. Further, since microglia are key initial immune responders to ethanol, and microglial depletion blunts many immune responses associated with ethanol, we hypothesized that microglial depletion would block the secretion of pro-inflammatory

MVs. Secretion of DAMPs such as HMGB1 in MVs utilizes an unconventional secretory mechanism that is dependent on PI3 kinase (PI3K)/Akt signaling (Dupont et al., 2011; Jiang et al., 2013; Nickel & Rabouille, 2009). Thus, we hypothesized that this pathway would be key for ethanol-induced secretion of pro-inflammatory MVs from microglia. In order to determine the role of MVs in pro-inflammatory activation by ethanol, we applied the stepwise approach recommended by the International Society on Extracellular Vesicles (ISEV, Figure 1) (Lotvall et al., 2014; Thery et al., 2018; Witwer et al., 2013). This involves analysis of MV contents, inhibition of MV secretion, and transfer of conditioned MVs. We utilized an organotypic brain slice culture (OBSC) model that contains all cell types *in situ*, undergoes functional synaptic maturation (Stoppini et al., 1991), and recapitulates inflammatory responses to ethanol found *in vivo* and in human postmortem brain (Coleman, Zou, & Crews, 2017; Coleman et al., 2020; Zou & Crews, 2014). Our findings indicate that MVs play an important role in regulating pro-inflammatory responses to ethanol.

## 2 | MATERIALS AND METHODS

### 2.1 | Materials

The colony-stimulating factor 1 receptor (CSF1R) inhibitor PLX3397 was obtained from MedChemExpress (Monmouth Junction, USA).

Imipramine (IMP) was purchased from Sigma (St. Louis, MO). Wortmannin (WORT) was obtained from Calbiochem (USA). The HMGB1 ELISA was obtained from Immuno-Biological Laboratories Co. International (Hamburg, Germany).

### 2.2 | Experimental design and rigor

This was an exploratory study with the overall experimental design as depicted in Figure 1. Briefly, OBSC were (A) treated with ethanol, and expression of microglial and pro-inflammatory genes as well as the number and size distribution of media MVs were measured. This was done with and without IMP, an inhibitor of MV secretion. In (B), experiments testing the effect of ethanol-induced MVs on pro-inflammatory and microglial genes are depicted. MVs were isolated from control or ethanol-treated slices and transferred to naïve OBSC. This was done with or without the HMGB1 inhibitor glycyrrhizin (GLY). Each ethanol or ethanol-MV-treated group was compared to a control or control-MV-exposed specimens treated for the same amount of time. All findings were repeated at least once to ensure reproducibility. All approaches for determining the role of MVs in ethanol neuroimmune induction were in accordance with guidelines for studying EV biology recommended by the ISEV (Lotvall et al., 2014; Thery et al., 2018; Witwer et al., 2013).

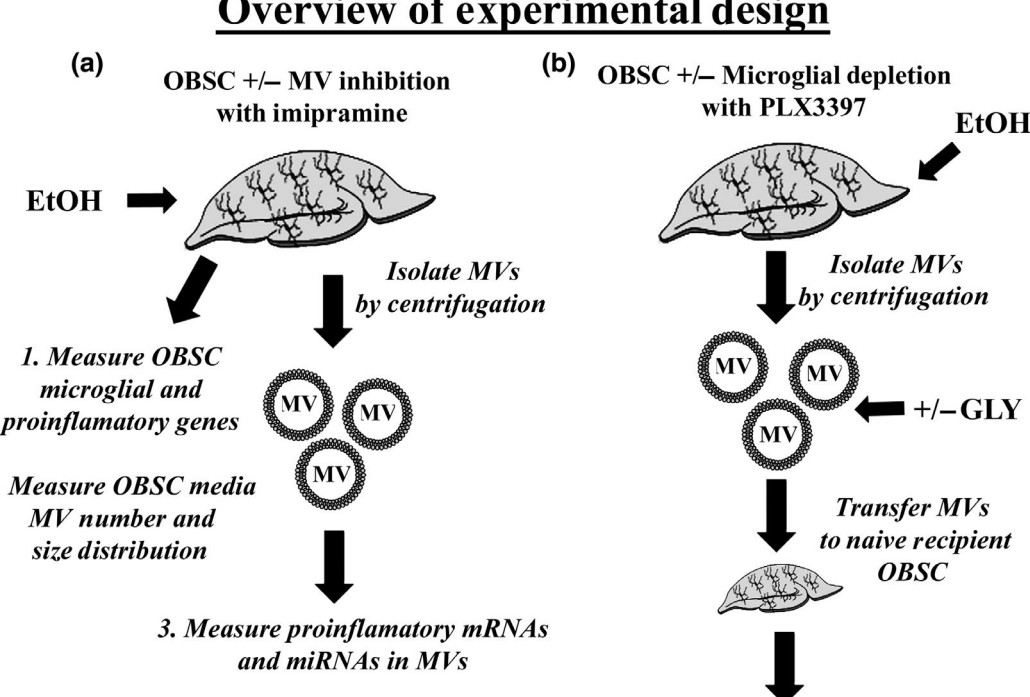

**Overview of experimental design**

**FIGURE 1** Overall experimental design. Primary organotypic brain slice culture (OBSC) was used to evaluate the role of extracellular microvesicles (MVs) on ethanol induction of pro-inflammatory signaling. (A) OBSC received ethanol ± MV secretion inhibitor imipramine with measurement of pro-inflammatory genes in OBSC. Media MV concentration was measured by nanoparticle tracking analysis. MVs were isolated, and pro-inflammatory mRNAs and miRNAs were measured. (b) OBSC ± microglial depletion with colony-stimulating factor 1 (CSFR1) antagonist PLX3387 were treated with ethanol. MVs from ethanol-treated OBSC were transferred to naïve OBSC ± the HMGB1 inhibitor glycyrrhizin (GLY) and pro-inflammatory cytokines measured in MV-treated OBSC

## 2.3 | Primary organotypic brain slice culture

Primary OBSCs were prepared from the hippocampal-entorhinal cortex formation of postnatal day 7 pups using previously established techniques of Stoppini et al. (1991) with modifications as we have described previously (Zou & Crews, 2005, 2006). Briefly, pregnant Sprague Dawley rat mothers were obtained from Charles River (Raleigh, NC, USA). Pregnant mothers were single housed. Neonates (an average of 6 per litter) at postnatal day 7 were decapitated, the brain was removed, and hippocampal-entorhinal complex dissected in Gey's buffer (Sigma-Aldrich, St. Louis, MO). Slices (~25/pup) were transversely cut with McIlwain tissue chopper at a thickness of 375 μm and placed onto a 30-mm diameter Millicell low height culture insert (Millipore, PICMORG50, St. Louis, MO), 10–13 slices/tissue insert. Slices from both sexes were pooled together. Slices were cultured with MEM containing 25 mM HEPES and Hank's salts, supplemented with 25% horse serum (HS) + 5.5 g/L glucose + 2 mM L-glutamine in a humidified 5% $CO_2$ incubator at 36.5°C for 7 days in vitro (DIV), followed by 4DIV in MEM + 12% HS, and then slices were cultured with MEM + 6% HS. Two replicates were performed for each experimental condition, with each experiment repeated at least twice. Serum-free N2-supplemented MEM was used to mix with MEM containing 25% HS throughout experiments. OBSC slices were incubated for a total of 14 days prior to treatment as described by Stoppini et al. during which OBSC stabilize as they continue to show functional maturation of synapses (Stoppini et al., 1991). OBSC slices are long-lived with no detectable cell death up to 42 days in culture in our previous report (Coleman et al., 2020).

## 2.4 | Ethanol treatment of OBSC

All ethanol treatment with the indicated concentrations occurred in a desiccator containing 300 ml water plus ethanol at the same concentration present in the culture media. For regular slice cultures, OBSC slices at 14DIV were exposed to ethanol (100 mM) for different durations. Though this is a high concentration of alcohol for nondependent individuals, patients with AUD reach these blood alcohol concentrations while being conscious and functional (Olson et al., 2013), and we have previously found this concentration *ex vivo* causes induction of immune genes that is similar to findings *in vivo* and in postmortem human AUD brain (Coleman et al., 2017; Crews et al., 2013). For ethanol treatment following microglial depletion, both control and microglia-depleted slices (after PLX3397 for 10DIV) were exposed to ethanol (100 mM) for 4 days in MEM + 6% HS. At the end of experiments, slices were removed for further analysis.

## 2.5 | Measurement of MV numbers and concentration by nanoparticle tracking analysis (NTA)

Media MV concentration was measured by NTA using a ParticleMetrix ZetaView® machine in the UNC Nanomedicine Core.

Medium samples were diluted 1:1,000 in filtered PBS (<0.02 μm). The background concentration of MVs measured in filtered PBS was subtracted from each sample to obtain final media MV concentrations.

## 2.6 | MV isolation and transfer experiments

The culture media were collected at the end of the experiments as indicated. For MV isolation, media were centrifuged at 300 g for 10 min and followed by 6,000× g for 10 min to remove cellular debris. The supernatant was then centrifuged at 21,000× g for 96 min at 4°C to pellet MVs, and pelleted MVs were used for different purposes (see below). For fluorescent labeling of MVs, MV pellets were labeled using PKH67 dye (Sigma-Aldrich, Cat# MIDI67-1KT), a fluorescent dye that labels the lipid membranes, according to the manufacturer's instructions with some modifications. Briefly, the MV pellet was resuspended in 200 μl of diluent C (provided by the manufacturer) and 6 μl of PKH67 was added. After incubation for 5 min at room temperature, labeling was stopped by addition of an equal volume of medium (200 μl) containing 25% horse serum for 1 min. The labeled MVs were then repelleted and washed once with PBS to remove any residual dye. The final labeled MV pellet was resuspended with MEM containing 6% HS and applied to new slice cultures. Cellular uptake of labeled MVs was observed using a Leica Stellaris 5 confocal microscope. For MV transfer experiments, pelleted MVs from control or ethanol-treated groups were resuspended in MEM + 6% HS. A ratio of MVs pelleted from two slice culture wells to one slice culture well was used with treatments of different durations. At the end of MV treatment, OBSC sections were removed for mRNA analysis by RT-PCR. To ascertain the role of non-MV components, media were collected from control and ethanol-treated OBSC and stored in a sterile manner in the incubator overnight to allow remaining ethanol to evaporate from the media as we have previously reported (Walter & Crews, 2017). MVs were removed from either control or ethanol-conditioned media by centrifugation. MV-depleted media, in a 1:1 dilution with fresh media to ensure appropriate cellular nutrition and media pH, were added to naïve OBSC, and pro-inflammatory gene expression was measured. For HMGB1 inhibition, MVs were resuspended in GLY (200 μM), a concentration we previously reported is effective at blunting ethanol-mediated induction of pro-inflammatory genes (Zou & Crews, 2014), for 30 min, then repelleted by centrifugation (21,000× g, 96 min) to remove GLY, and resuspended in media for addition to naïve OBSC.

## 2.7 | Measurement of immune mRNAs in MVs

The pelleted MVs were used for purification of total RNA by using miRNAeasy Kit (Qiagen Inc., CA, USA) according to the manufacturer's protocol. Briefly, MVs from 2 to 3 ml of culture media were dissolved in total of 700 μl Trizol. After centrifugation at 12,000× g for 15 min, supernatant was removed for total RNA purification. A total

of 28 μl RNA was collected and concentration of RNA was measured by Nanodrop™. The expression of mRNA level was determined using RT-PCR (see below).

## 2.8 | Reverse transcription and RT-PCR

For each specific experiment, the slices were removed at the end of the experiment, rinsed with cold PBS, and followed by total RNA purification using miRNeasy Kit (Qiagen, CA). The total amount of RNA was quantified by Nanodrop™. For reverse transcription, either 200 ng of RNA from MVs or 2 μg of RNA from slices was used to synthesize the first strand of cDNA using random primers (Invitrogen, ThermoFischer, MA, USA) and reverse transcriptase Moloney murine leukemia virus (Invitrogen). After a 1:2 dilution with water, 2 μl of the first strand cDNA solution was used for RT-PCR. The primer sequences for real-time RT-PCR are shown in Table 1. Primer sequences were validated using the NCBI Primer-BLAST, with only primers that targeted the desired mRNA and had single-peak melt curves (showing amplification of a single product) being used. Further, the sizes of the expected PCR products for TNF-α, C1q, and TRAIL in MVs were confirmed by running the products on polyacrylamide gels. SYBER Green Supermix (AB system, UK) was used as a RT-PCR solution. The real-time RT-PCR was run with initial activation for 10 min at 95°C and followed by 40 cycles of denaturation (95°C, 40 s), annealing (58°C, 45 s), and extension (72°C, 40 s). The threshold cycle ($C_T$) of each target product was determined and normalized to a reference housekeeping gene. For all analyses in OBSC tissue, β-actin was used as the housekeeping gene. For measurement of mRNAs in MVs, 18S was used as the housekeeping gene since actin mRNA was not detected in the vesicles. Difference in $C_T$ values ($\Delta C_T$) of two genes was calculated [difference = $2^{-(C_T \text{ of target genes} - C_T \text{ of β-actin})} = 2^{-\Delta C_T}$], and the result was expressed as the percentage compared to control.

## 2.9 | Microglial depletion in OBSC

For microglial depletion experiments, OBSC slices at 4DIV were treated with CSF1R inhibitor PLX3397 (1 μM, Active Biochem, Hong Kong) for 7 days in regular culture medium (MEM containing 25% HS), followed by 3DIV in MEM + 12% HS to deplete microglia. We used 1μM PLX3397 since we have previously reported that this successfully depletes >90% of microglia in our *ex vivo* slice culture model (Coleman et al., 2020). At the end of PLX3397 treatment, slices were either removed for analysis or followed by ethanol treatments for 4DIV in MEM + 6%HS without PLX3397.

## 2.10 | BV2 microglial cultures

BV2 microglia (from ICLC #ATL03001; Genoa, Italy) were maintained in culture in standard cell culture conditions as we and others have described previously (Coleman, Zou, Qin, et al., 2017; Lawrimore et al., 2019). Briefly, $3 \times 10^5$ cells were plated per well in 6-well culture plates in DMEM with 10% FBS, 1× glutaMAX, and 1× penicillin/streptomycin antibiotic (Life Technologies). Cells adhered overnight and were treated with ethanol (100 mM) the following morning.

## 2.11 | Study design, sample size, statistical methods

For each experiment, 9–10 randomly pooled slices per group with two technical replicates were used. Sample size was set using our

**TABLE 1** Primers for RT-PCR analyses

| Gene | Forward (5′-3′) | Reverse (5′-3′) |
|---|---|---|
| TNF-α | AGCCCTGGTATGAGCCCATGTA | CCGGACTCCGTGATGTCTAAG |
| IL-1β | TTGTGCAAGTGTCTGAAGCA | TGTCAGCCTCAAAGAACAGG |
| Tmem119 | AGTCGAACGGTCTAACAGGG | AAGAGGCTGAAGAACCCTCA |
| P2RY12 | GATTCTCACCAACAGGAGGC | ACAGAGTGTTCTCGGCATTG |
| CX3CR1 | CTTCTTCCTCTTCTGGACGC | CCTCGCTTGTGTAGTGAGTC |
| C1q | AGCTTTCTCAGCTATTCGGC | GGAGGAGGACACGATAGACA |
| C3R | TCGAGGAACAAAACCCTTCG | CCCATCACCAGTATTCACCTC |
| TRAIL | CTCGGTCATATCAGT GGTGC | GTTCTGTCAGGTTCCGTGTT |
| MARCKS | TCTGTGGTGGCCTCTCAATA | TTGGCCAACTAGGGGTTTTC |
| CX3CL1 | GTCGACTCCTCTAGACTCCC | CTATGGGTCTCTTGGCTCCT |
| IL-10 | CAAAGGTGTCTACAAGGCCA | CAAGGAGTTGCTCCCGTTAG |
| IL-4 | TGCACCGAGATGTTTGTACC | CGAGAACCCCAGACTTGTTC |
| IL-1α | CTACTTCACATCCGCAGCTT | CCATGCGAGTGACTTAGGAC |
| TREM2 | AAGCTTCTTACAGCCAGCAT | GTAGCAGAACAGAAGTCTTGGT |
| β-Actin | CTACAATGAGCTGCGTGT | CAGGTCCAGACGCAGGAT |
| 18S | CGGGGAATCAGGGTTCGATT | TCGGGAGTGGGTAATTTGCG |

historic effect sizes. The mean (±SEM) in each box plot is the average value across all experimental replicates. For each ethanol treatment experiment, ethanol-treated groups were compared to untreated control slices, with two to eight experimental replicates. As we have reported previously, the genes measured did not change in untreated slices from 0 hr to the longest treatment duration (96 hr) (Coleman et al., 2020). Therefore, control slices at the 96-hr time point were used, and groups were analyzed by one-way ANOVA with Dunnett's multiple comparisons test. The Brown-Forsythe test was used when the standard deviations were calculated to be significantly different across groups (GraphPad Prism). For MV transfer experiments, slices treated with ethanol-conditioned MVs (i.e., MVs from ethanol-treated slice cultures) were compared to slices treated with control-conditioned MVs (MVs from control, untreated slices) for the same duration, with three to eight experimental replicates and analyzed by two-way ANOVA with Sidak's multiple comparisons test. A $p$ value less than 0.05 was considered significant. This is consistent with the core set of standards for rigorous reporting issued by the NIH (Landis et al., 2012).

## 3 | RESULTS

### 3.1 | Ethanol induces pro-inflammatory genes in OBSC and secretion of pro-inflammatory mRNAs in MVs

Microglia are considered primary immunoregulatory cells in brain. Microglial activation state has been historically determined by morphology. Ethanol causes hyper-ramification of microglia along with induction of pro-inflammatory genes (He & Crews, 2008; Qin & Crews, 2012; Walter & Crews, 2017). Recently, unbiased sequencing studies have found that microglial activation in disease models is associated with changes in expression of genes that are specific to microglia such as Tmem119 and P2RY12 (Deczkowska et al., 2018). We used OBSC as a model to measure temporal changes in microglial-associated pro-inflammatory, activation, and functional genes in response to ethanol. OBSC were treated with ethanol (100 mM), a concentration that causes similar pro-inflammatory gene induction found in postmortem human AUD brain, and does not cause neuronal death (Coleman et al., 2020).

Ethanol caused a transient induction of the expression of pro-inflammatory genes TNF-α (Figure 2a, ANOVA main effect of treatment $F_{4, 25} = 10.44$, $p < 0.0001$) and IL-1β ($F_{4, 25} = 2.981$, $p = 0.039$). Post hoc analyses found TNF-α was significantly increased by 2 hr of ethanol treatment ($^{††}p < 0.01$, Dunnett's multiple comparisons test) and IL-1β increases reached statistical significance at 24 hr ($^{†}p < 0.05$). We reported previously that microglial depletion *in vivo* and in OBSC removes TNF-α, consistent with microglial localization (Coleman et al., 2020; Walter & Crews, 2017). Ethanol also altered genes associated with different microglial activation states. Tmem119 and P2RY12 are considered markers of "homeostatic" microglia (Galatro et al., 2017; Haage et al., 2019; Kaiser & Feng, 2019;

Satoh et al., 2016). However, P2RY12, a purinergic receptor, is also localized in both morphologically resting and activated microglia (Walker et al., 2020). Ethanol caused a progressive induction of Tmem119 mRNA (Figure 2b, $F_{4, 11} = 3.417$, $p = 0.048$) that reached 1.9-fold at 96 hr, and increased P2RY12 ($F_{4, 12} = 10.48$, $p = 0.0007$), reaching a 2.50-fold induction at 48 hr. Interestingly, ethanol also increased the microglial inhibitor receptor CX3CR1 ($F_{5, 18} = 6.424$, $p = 0.0014$, Brown-Forsythe test). CX3CR1 is the receptor for fractalkine (CX3CL1), a neuronal inhibitory signal for microglia. Thus, ethanol causes a unique activation state in microglia as determined by microglial gene expression. We next assessed the temporal effect of ethanol on microglial functional genes C1q (synaptic pruning), C3R (synaptic regulation), and TREM2 (phagocytosis). A significant main effect of ethanol to increase C1q ($F_{4, 10} = 26.81$ $p < 0.0001$), C3R ($F_{4, 10} = 4.521$, $p = 0.024$, Brown-Forsythe test), and TREM2 ($F_{4, 10} = 2.72$, $p = 0.09$) was also found (Figure 2c). Thus, these results find that ethanol causes traditional pro-inflammatory gene induction, with a complex induction of microglial activation and functional genes.

We have reported previously that ethanol alters the content of MVs that are secreted from brain tissue, with increases in HMGB1 protein and multiple pro-inflammatory miRNAs including let-7 (Coleman, Zou, & Crews, 2017). MVs could serve as diagnostic biomarkers and might also mediate immune pathology. Therefore, we assessed the content of immune-modulating mRNAs in MVs in response to ethanol. MVs (~0.1–1 μm diameter) were isolated from OBSC media by centrifugation, and the levels of immune-modulating mRNAs were measured by RT-PCR. Ethanol caused a significant increase in several pro-inflammatory mRNAs in MVs that we also have found were increased in OBSC slice tissue with ethanol, as summarized in Figure 3. This includes TNF-α (8-fold), HMGB1 (4.7-fold), and TRAIL (8.2-fold). Interestingly, none of the mRNAs for IL-1β, IL-6, or IL-4 were detected in MVs. Ethanol also caused trends toward increased expression of mRNAs for anti-inflammatory IL-10 (twofold) and CX3CL1 (fourfold). C1q was also increased in MVs after ethanol (fivefold). Increases in MV mRNA for TNF-α, TRAIL, and C1q mirror findings in OBSC tissue, though the magnitude of induction is greater in MVs. However, it is unclear if these mRNAs are transferred in a functional manner to recipient cells or if they are simply markers of immune activation.

### 3.2 | Inhibition of MV secretion blocks ethanol-induced pro-inflammatory gene activation

MVs have been found to promote intercellular communication in other disease settings (Raeven et al., 2018; Tkach & Thery, 2016). Therefore, we investigated whether MVs play a role in pro-inflammatory responses to ethanol. Secretion of MVs requires membrane reorganization that involves changes in membrane fluidity. This is mediated in part by hydrolysis of sphingomyelin to ceramide by acid sphingomyelinase (aSMase), with aSMase inhibition reducing MV secretion (Bianco et al., 2009; Catalano & O'Driscoll, 2020).

Imipramine (IMP) is a tricyclic antidepressant that also acts as an inhibitor of aSMase, reducing the secretion of MVs (Bianco et al., 2009). Therefore, we used IMP to determine the effect of aSMase-mediated MV secretion on ethanol-mediated induction of pro-inflammatory genes. We first measured the concentration of MVs in OBSC media using NTA. OBSCs were treated for 24 hr since this was the earliest time point with maximal induction of TNF-α and IL-1β (Figure 2a). We tested IMP concentrations 1–10 μM, previously reported to cause aSMase inhibition and to reduce MV secretion (Bianco et al., 2009; Catalano & O'Driscoll, 2020). Ethanol

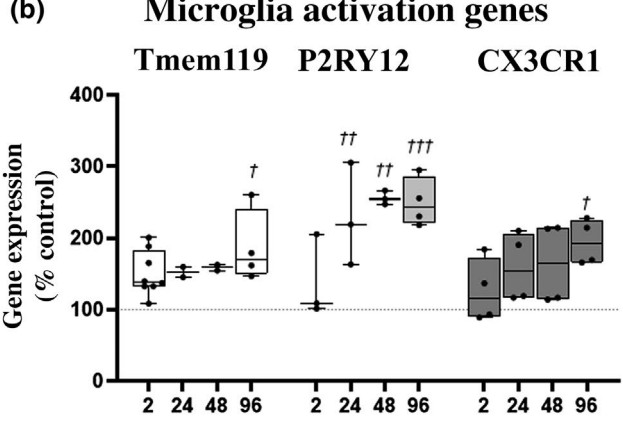

**(a)**

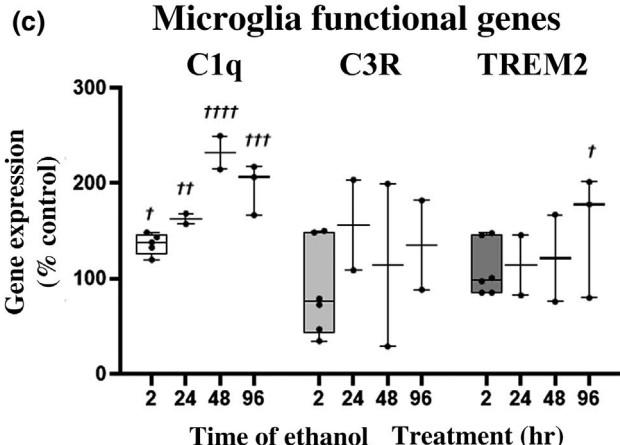

**(b)**

**(c)**

**FIGURE 2** Ethanol induction of pro-inflammatory and microglial modulating genes. Primary organotypic brain slice culture (OBSC) was treated with ethanol (100 mM) for different durations and gene changes assessed by RT-PCR and compared to controls. Genes of interest were normalized to the housekeeping gene β-actin. Each data point represents an experimental replicate with $N = 2$–8 experimental replicates/group. Each experimental replicate is the average value of two technical replicates with nine pooled slices/well. (a) Temporal induction of TNF-α and IL-1β by ethanol. A main effect of ethanol treatment on TNF-α (ANOVA $F_{4, 25} = 10.44$, $p < 0.0001$) and IL-1β ($F_{4, 25} = 2.981$, $p = 0.039$) gene expression was observed with post hoc analysis finding TNF-α induction reached significance by 2 hr and IL-1β reached a significant twofold induction by 24 hr. (b) Alteration in microglial activation genes. A significant main effect of ethanol treatment on Tmem119 was found ($F_{4, 11} = 3.417$, $p = 0.048$) with a 1.9-fold increase at 96 hr ($^{†}p < 0.05$, Dunnett's multiple comparisons test). A main effect of ethanol on P2RY12 was also observed ($F_{4, 12} = 10.48$, $p = 0.0007$), with a progressive increase reaching a 2.50-fold induction at 48 hr. CX3CR1 showed a progressive increase ($F_{5, 18} = 6.424$, $p = 0.0014$, Brown-Forsythe test). (c) Microglial functional genes. Ethanol progressively induced C1q ($F_{4, 10} = 26.81$, $p < 0.0001$). An increase in C3R ($F_{4, 10} = 4.521$, $p = 0.024$, Brown-Forsythe test) and a trend toward significant reduction in TREM 2 ($F_{4, 10} = 2.72$, $p = 0.09$) were found. $^{†}p < 0.05$, $^{††}p < 0.01$, $^{†††}p < 0.001$, $^{††††}p < 0.0001$ versus Control, Dunnett's multiple comparison test

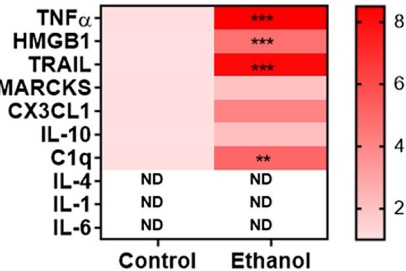

**FIGURE 3** Induction of immune-modulating mRNAs in MVs. Primary organotypic brain slice culture (OBSC) was treated with ethanol (100 μM) for 96 hr. MVs were isolated from media by centrifugation, total RNA isolated, and gene expression measured by RT-PCR. Genes of interest were normalized to the housekeeping gene 18S. Heat map depiction of expression of immune-modulating mRNAs. Multiple $t$-tests were performed with FDR correction for multiple comparisons ($Q = 5\%$). $N = 2$–4 wells/group $^{**}p < 0.01$, $^{***}p < 0.001$ and $q < 0.05$. ND, not detected [Color figure can be viewed at wileyonlinelibrary.com]

and IMP (1 μM) each caused a similar reduction in the concentration of media MVs (~30%, Figure 4a). The mechanism of the reduction of MVs by ethanol is not clear but might involve increased uptake of MVs from target cells (Mulcahy et al., 2014) as microglial depletion blunts this effect (see Figure 6c). The combination of ethanol and IMP resulted in a reduction of media MV concentration (~53%) down near to the level of medium supernatant (SUPN) samples in which MVs were depleted by centrifugation. Analysis of the MV size distribution by NTA found a robust reduction

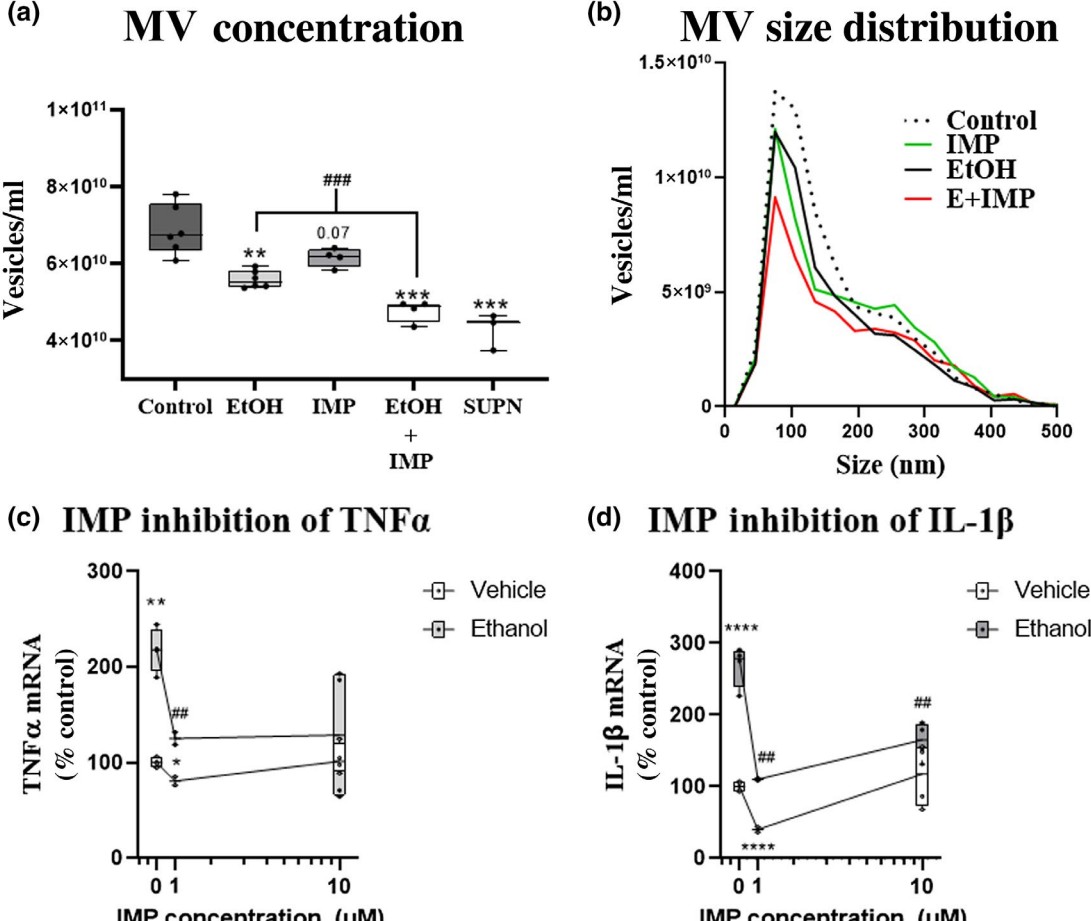

**FIGURE 4** Inhibition of microvesicle (MV) secretion blocks pro-inflammatory gene induction by ethanol. Primary organotypic brain slice culture (OBSC) was treated with either ethanol (100 mM), imipramine (IMP, 1 µM), or both (E+IMP) for 24 hr. (a) MVs were isolated from OBSC media by centrifugation. MV concentration was measured using nanoparticle tracking analysis (NTA). Ethanol and IMP alone each caused a ~30% reduction in MV concentration. EtOH+IMP reduced MV concentration by 53% near the level of MV-depleted supernatant (SUPN). $N = 5$-6/group $**p < 0.01$, $***p < 0.001$ versus control, $###p < 0.001$ versus EtOH. (b) MV size distribution of the same samples was determined by NTA. The average curves for each group are shown. EtOH+IMP resulted in a reduction in 50–350 nm MVs. $N = 3$–6 wells/group. (c,d) The effect of IMP on pro-inflammatory gene induction by ethanol. IMP reduced ethanol induction of (c) TNF-α from 2.3-fold and (d) IL-1β from 2.5-fold to control levels. Genes of interest were normalized to the housekeeping gene β-actin. $*p < 0.05$ versus control, $#p < 0.05$ versus EtOH alone. $N = 1$–2 experimental replicates per group, two technical replicates/experiment, nine pooled slices each [Color figure can be viewed at wileyonlinelibrary.com]

of MVs in the ~50–350 nm diameter size range, which includes larger exosomes (50–150 nm) and MVs (50–1,000 nm) (van Niel et al., 2018). To investigate the role of MV secretion of pro-inflammatory activation by ethanol (24 hr), we measured TNF-α and IL-1β in the presence of IMP (1 or 10 µM). IMP blocked ethanol induction of pro-inflammatory genes TNF-α and IL-1β (Figure 4c,d, respectively, $##p < 0.01$ vs. EtOH alone), with increased variability seen with 10 µM IMP suggesting possible off-target effects at higher concentration. Thus, IMP blocked MV release and blunted pro-inflammatory gene induction by ethanol, consistent with MVs promoting ethanol-induced pro-inflammatory activation in brain. However, IMP also has effects on several neurotransmitter systems which could promote its anti-inflammatory effects. Therefore, we next employed ethanol-conditioned MV transfer

experiments to further clarify their role in ethanol induction of pro-inflammatory responses.

## 3.3 | Ethanol-secreted MVs reproduce pro-inflammatory gene activation in OBSC seen with ethanol

Since inhibition of MV release blunted pro-inflammatory responses to ethanol, we next asked if MVs secreted in response to ethanol can directly induce the pro-inflammatory activation and microglial gene changes seen with ethanol alone. Ethanol-conditioned MVs, or EtOH-MVs, were isolated from ethanol-treated OBSC media by differential centrifugation to obtain 100–500 nm diameter MVs. This

same size range was found to be reduced by IMP with EtOH treatment in Figure 4b. The MV isolation results in the removal of residual media, meaning the observed effects are mediated by MVs or other pelleted proteins or lipids rather than any remaining ethanol. We assessed multiple dilutions of MVs (data not shown). We found that 96 min of centrifugation at 21,000× *g* isolates approximately 30%

of the total media MVs (see Figure 4 SUPN levels), therefore a 2:1 well ratio chosen (i.e., MVs isolated from two wells transferred to one naïve well). We first labeled isolated MVs with PKH67 and imaged MV uptake in recipient brain slices (Figure 5a). Cell nuclei were stained blue with DAPI and the MV fluorescent dye pictured as red for optimal color contrast. Stained MVs are clearly seen within the

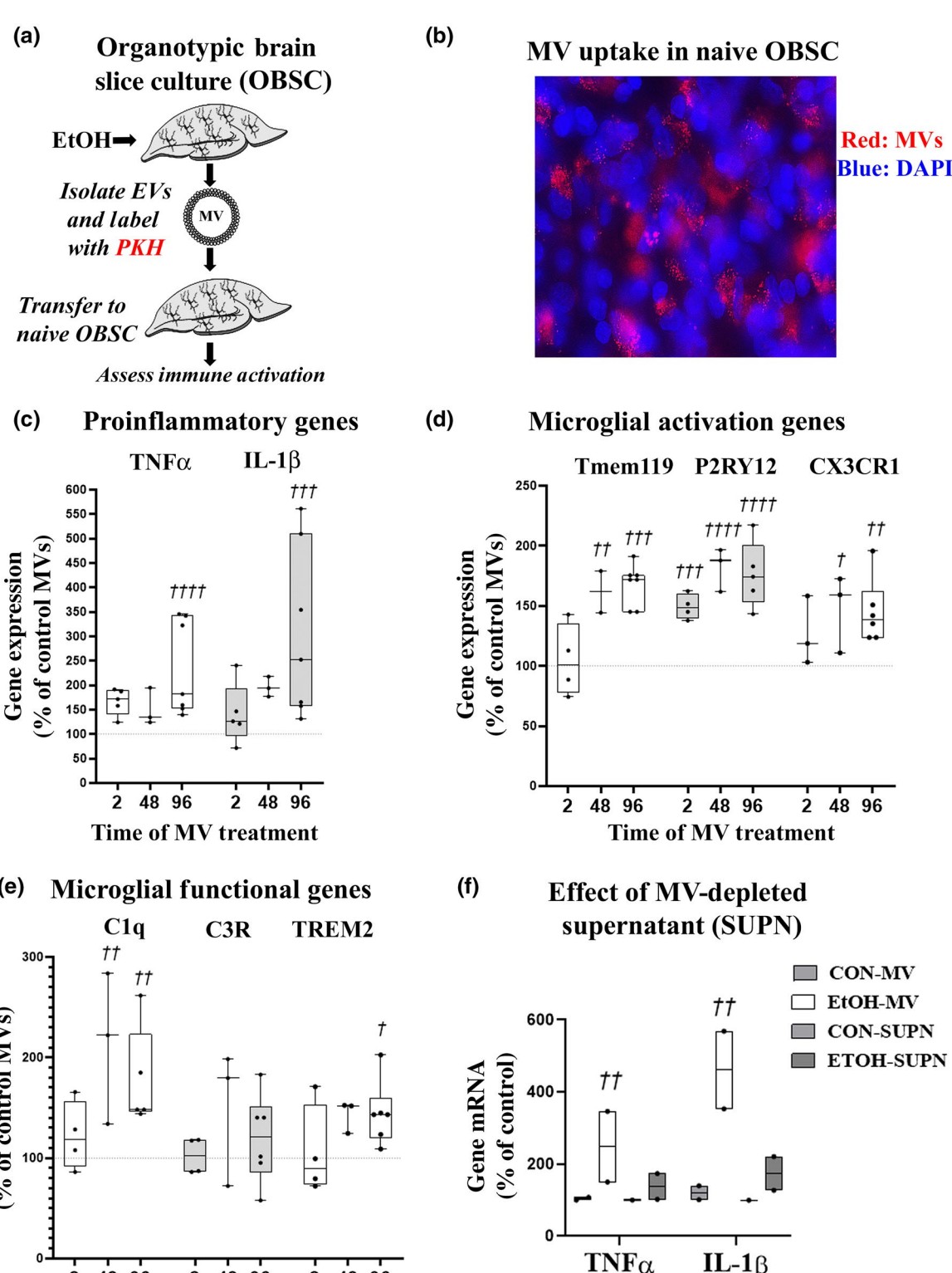

**FIGURE 5** Ethanol-conditioned microvesicles (MVs) reproduce pro-inflammatory gene induction in naïve slices. (a,b) Confirmation of MV transfer. Primary organotypic brain slice cultures (OBSC) were treated with ethanol (100 mM, 24 hr). MVs were isolated by centrifugation and labeled with fluorescent membrane dye PKH67. MVs were washed with PBS to remove residual dye, repelleted and then transferred to naïve OBSC. Co-immunofluorescence with DAPI shows MVs in the cytoplasm but not in the nucleus. (c–e) MVs from time-matched control or ethanol-conditioned MVs were transferred to naïve OBSC for different durations. Ethanol-conditioned MV (EtOH-MV) gene induction is presented as percent change relative to control-conditioned MV (control-MV) effects. Each data point represents an experimental replicate which is the average value of two technical replicates consisting of nine pooled slices/well. Genes of interest were normalized to the housekeeping gene β-actin. (c) EtOH-MVs caused temporal induction of TNF-α and IL-1β. A significant main effect of ethanol on TNF-α was seen (two-way ANOVA$_{Treatment}F_{1, 25}$ = 20.39, $p$ = 0.0001) that reached a 2.3-fold significant increase at 96 hr. A main effect of ethanol on IL-1β was also seen ($F_{1, 25}$ = 10.95, $p$ = 0.0028) reaching a threefold increase at 96 hr. [†††]$p < 0.001$, [††††]$p < 0.0001$-Sidak's multiple comparison test. (d) EtOH-MVs altered microglial activation genes. A main effect of ethanol was seen on Tmem119 ($F_{1, 20}$ = 43.6, $p < 0.0001$) reaching 1.7-fold induction at 96 hr and P2R12 ($F_{1, 18}$ = 112.3, $p < 0.0001$). A main effect of ethanol on CX3CR1 was observed ($F_{1, 20}$ = 25.42, $p < 0.0001$). (e) Microglial functional genes were altered by EtOH-EV: Main effect of ethanol on C1q ($F_{1, 19}$ = 22.69, $p$ = 0.0001), a trend toward an effect on C3R ($F_{1, 20}$ = 3.482, $p$ = 0.08), and a main effect of ethanol on TREM2 ($F_{1, 20}$ = 9.830, $p$ = 0.0052). [†]$p < 0.05$, [††]$p < 0.01$, control-MV versus EtOH-MV, Sidak's multiple comparison's test. (f) Naïve OBSCs were treated with either control-MVs (CON-MV), ethanol-conditioned MVs (EtOH-MV), control-MV-depleted medium supernatant (CON-SUPN) or ethanol-treated MV-depleted media (ETOH-SUPN). EtOH-MVs induced TNF-α and IL-1β responses in naïve OBSCs, while MV-depleted media did not [Color figure can be viewed at wileyonlinelibrary.com]

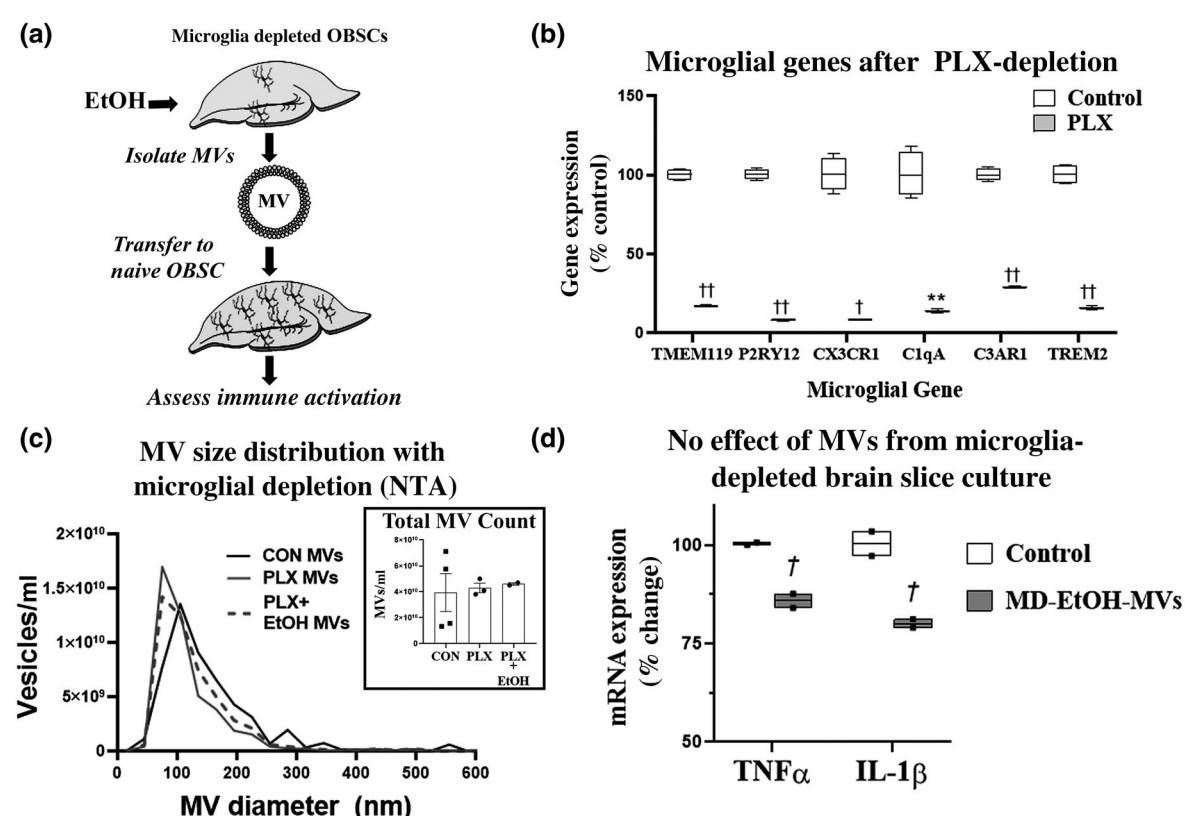

**FIGURE 6** Microglial depletion prior to ethanol treatment blocks microvesicle (MV) induction of pro-inflammatory genes. (a) Experimental design. Microglia were depleted from primary organotypic brain slice culture (OBSC) with PLX3397 prior to nanoparticle tracking analysis (NTA) of MVs, ethanol treatment, and MV transfer. (b) Loss of microglia-specific genes after microglial depletion. Genes of interest were normalized to the housekeeping gene β-actin. **$p < 0.05$, [†]$p < 0.001$, [††]$p < 0.0001$ versus control. (c) NTA of MVs isolated from microglia-depleted and control slices ± EtOH treatment. Average curves for each group are shown. MV size distribution and total number of isolated MVs were not significantly impacted by microglial depletion. (d) MVs isolated from microglia-depleted OBSC with microglia that were treated with ethanol (MD-EtOH-MVs) were administered to naïve OBSC for 96 hr. MD-EtOH-MVs did not induce TNF-α and IL-1β, rather they were slightly reduced. [†]$p < 0.05$ versus control

cytoplasm of cells and not in the nucleus, consistent with cellular uptake of MVs (Figure 5b).

We next performed conditioned MV transfers to naïve, un-treated recipient OBSC slice cultures. Since MV transfer increases

the total number of media MVs from baseline levels, all findings from EtOH-MVs were compared to MVs from DIV-matched con-trol OBSC (control-MVs) to account for any potential nonspecific effects caused by an increase in total media MVs. We therefore

compared the effects of control-MVs and EtOH-MVs on OBSCs treated for the same duration on: pro-inflammatory genes, microglial activation genes, and microglial functional genes. EtOH-MVs caused similar changes in gene induction seen with ethanol treatment alone. Similar to findings with ethanol (Figure 2a), EtOH-MVs induced the expression of TNF-α (twofold) in naïve OBSC slices within 2 hr (Figure 5c, two-way ANOVA$_{Treatment}$ $F_{1, 25} = 20.39$, $p = 0.0001$ with Sidak's multiple comparison's test, $^{†††}p < 0.001$, $^{††††}p < 0.0001$, control-MV versus EtOH-EV). Likewise, a significant main effect of ethanol on induction in IL-1β was found (Figure 5c, ANOVA$_{Treatment}$ $F_{1, 25} = 10.95$, $p = 0.0028$) that reached statistical significance at 72 and 96 hr of exposure (Sidak's multiple comparisons test $^{†††}p < 0.001$, $^{††††}p < 0.0001$). Both the magnitude and time course of pro-inflammatory gene induction were similar to effects with ethanol treatment on OBSC. EtOH-MVs also caused similar temporal changes in microglial activation genes as seen with ethanol. Similar to ethanol treatment (Figure 2b), a main effect of EtOH-MVs on the microglial surface marker Tmem119 was found (Figure 5d, ANOVA$_{Treatment}$ $F_{1, 20} = 43.6$, $p < 0.0001$) that reached nearly a twofold increase at 96 hr of exposure. A main effect of EtOH-MVs on P2R12 gene induction was also seen ($F_{1, 18} = 112.3$, $p < 0.0001$), reaching a ~1.5-fold increase within 2 hr ($^{†††}p < 0.001$, Sidak's multiple comparison test). EtOH-MVs also induced the expression of CX3CR1 ($F_{1, 20} = 25.42$, $p < 0.0001$), reaching ~1.5-fold induction within 48 hr. The pattern of these microglial changes was very similar to those seen with ethanol treatment (Figure 2b). Microglial functional genes C1q (ANOVA$_{Treatment}$ $F_{1, 19} = 22.69$, $p = 0.0001$), C3R ($F_{1, 20} = 3.482$, $p = 0.08$), and TREM2 ($F_{1, 20} = 9.830$, $p = 0.0052$) also followed similar temporal changes with EtOH-MVs (Figure 5e) as with EtOH (Figure 2c). These findings indicate that EtOH-MVs produce similar temporal induction of pro-inflammatory genes and microglial activation and function genes as ethanol, though there is no ethanol present in MV preparations. Next, to gain insight into whether changes induced by MVs are specific to the MV compartment, we compared the effect of EtOH-MVs with MV-depleted media. MV-depleted media from ethanol-treated OBSCs (EtOH-SUPN) did not induce TNF-α or IL-1β above the levels of control-SUPN in naïve OBSC, suggesting factors secreted in MVs are responsible for ethanol induction of pro-inflammatory genes and microglial activation.

### 3.4 | Pharmacological removal of microglia prevents pro-inflammatory actions of ethanol-MVs

As findings implicate MVs as novel mediators of ethanol-induced pro-inflammatory activation and microglial regulation, we next wondered which cell types are involved in the production of these pro-inflammatory MVs. Microglia are considered the first responders to immune activation in brain. Therefore, we hypothesized that microglia are critical to the production of pro-inflammatory MVs. Microglia can be depleted both *in vivo* and *ex vivo* with pharmacological inhibition of CSF1R, with no detectable nonmicroglial cell death

or behavioral dysfunction (Coleman et al., 2020; Han et al., 2017; Hughes & Bergles, 2014; Rice et al., 2015; Varvel et al., 2012; Walter & Crews, 2017). We have reported recently that 7 days of treatment with the CSF1R antagonist PLX3397 (1 μM) safely removes >90% of microglia from OBSC and blocks pro-inflammatory gene induction with ethanol (Coleman et al., 2020). We, therefore, assessed whether depletion of microglia would prevent the production of pro-inflammatory MVs by ethanol (Figure 6a). We confirmed that PLX3397 removed microglia using RT-PCR for microglia-specific genes. A panel of six different microglial genes (Tmem119, P2RY12, CX3CR1, C1qA, C3ar1, and TREM2) were each profoundly reduced by PLX3397 treatment (Figure 6b). We then assessed if microglial depletion altered the size distribution and number of MVs. MVs were isolated by centrifugation and assessed by NTA. Depletion of microglia with PLX3397 had no impact on MV number or size distribution (Figure 6c). Further, ethanol did not reduce MV concentration after microglial depletion, suggesting ethanol-induced reductions in media MVs seen in OBSC with microglia (Figure 4a) may be due to increased microglial engulfment of MVs in the presence of ethanol. However, microglial depletion prior to ethanol treatment prevented the formation of pro-inflammatory MVs. MVs isolated from PLX3397 OBSC treated with ethanol did not induce the expression of either TNF-α or IL-1β in naïve recipient OBSC (Figure 6d). Therefore, microglia are required for the production of pro-inflammatory EtOH-MVs.

### 3.5 | HMGB1 promotes activity of ethanol-MVs through PI3K-mediated secretion from microglia

Pro-inflammatory MVs that are induced by ethanol carry an assortment of protein, lipid, and nucleic acid cargo. We and others have reported previously that TLR activation is a key feature of alcohol-induced inflammatory responses (Crews, Lawrimore, et al., 2017; Montesinos et al., 2016). Further, we have reported that endogenous TLR ligands such as HMGB1 and let-7b are involved in TLR activation with ethanol (Coleman, Zou, & Crews, 2017; Crews et al., 2013). HMGB1 is a critical regulator of TLR signaling, acting as an agonist for TLR2 and TLR4, and being required for function of endosomal TLRs such as TLRs 3 and 7–9 (Bianchi, 2009; Castiglioni et al., 2011; Yanai et al., 2009). We have found that HMGB1 is present in MVs in the setting of alcohol exposure and other illnesses (Coleman Jr. et al., 2018; Coleman, Zou, & Crews, 2017). Therefore, we hypothesized that inhibition of HMGB1 in MVs would prevent their pro-inflammatory activity. MVs were isolated from either control or ethanol-treated OBSCs and incubated with the HMGB1 inhibitor GLY prior to their transfer to naïve OBSC. GLY binds directly to both the A and B boxes of HMGB1 ($K_D$ ~ 150 μM) (Mollica et al., 2007) and has been effective at reducing inflammation in a variety of preclinical models (Musumeci et al., 2014). After incubation with GLY (200 μM), MVs were repelleted and resuspended in media to remove any residual GLY that could have direct effects on the recipient OBSC aside from the neutralization of MV-associated HMGB1. As above, EtOH-MVs induced the expression of TNF-α and IL-1β in naïve OBSC.

Neutralization of HMGB1 in MVs with GLY prior to transfer abolished the pro-inflammatory activity of EtOH-MVs.

As we find HMGB1 and microglia are involved in the pro-inflammatory actions of MVs, we assessed the mechanism of HMGB1 secretion of microglia using cultured BV2 microglia. Microglia were treated with ethanol for 24 hr, followed by isolation of MVs of different sizes: MVs and larger exosomes (50–500 nm) or exosomes (<50 nm) by centrifugation which allows for separation based on size. Ethanol caused an increase in HMGB1 protein in media MVs, but not exosomes (Figure 7b). Thus, HMGB1 localization seems to involve intracellular trafficking to MVs but not larger exosomes. HMGB1 secretion in response to lipopolysaccharide from peripheral monocytes has been reported to involve a PI3K-mediated mechanism that is thought to involve unconventional secretory autophagy (Dupont et al., 2011; Oh et al., 2009). For this reason, we investigated whether PI3K inhibition with WORT at similar concentrations used by Oh et al. (2009) (200 nM) would reduce ethanol-induced HMGB1 secretion. We found that 100 nM caused a slight reduction in medium HMGB1 (not shown), while WORT (200 nM) completely blocked secretion of HMGB1 in MVs in response to ethanol, returning MV HMGB1 concentration to control levels (Figure 7b). Thus, these findings indicate that HMGB1 in MVs plays a key role in driving pro-inflammatory responses associated with ethanol, and that PI3K is involved in MV secretion of HMGB1 secretion from microglia.

## 4 | DISCUSSION

This work implicates a role for MV signaling in ethanol-mediated induction of pro-inflammatory activation. We found that ethanol induces pro-inflammatory cytokines (TNF-α and IL-1β) and microglial activation gene induction in a manner consistent with pro-inflammatory activation. Ethanol changed the cargo of MVs, and inhibition of MV secretion with IMP blocked pro-inflammatory gene induction due to ethanol. Transfer of ethanol-conditioned MVs to naïve OBSCs mimicked the magnitude and temporal regulation of pro-inflammatory gene induction seen with ethanol as well as much of microglial activation and functional gene changes. Microglia play a key role in this cellular communication, as depletion of microglia prior to ethanol treatment abolished the pro-inflammatory effects of ethanol-MVs. Further, DAMPs localized to MVs seem to play a key role, as inhibition of the TLR modulator HMGB1 also prevented inflammatory responses due to EtOH-MVs. HMGB1 is considered a sentinel of TLR signaling, as it is a direct agonist for TLR2 and TLR4, and is also required for proper functioning of endosomal TLRs (i.e., TLRs 3, 7, and 9) (Castiglioni et al., 2011; Yanai et al., 2009). We have reported previously that HMGB1 acts as a chaperone for the endogenous ligand for TLR7, miRNA let-7b (Coleman, Zou, & Crews, 2017), and could act in a similar manner for other nucleic acid DAMPs. This secretion of HMGB1 in MVs from microglia involved PI3K activation. This is consistent with findings in peripheral monocytes, where HMGB1 secretion has been found to utilize an unconventional autophagy-mediated secretory system that may require PI3K

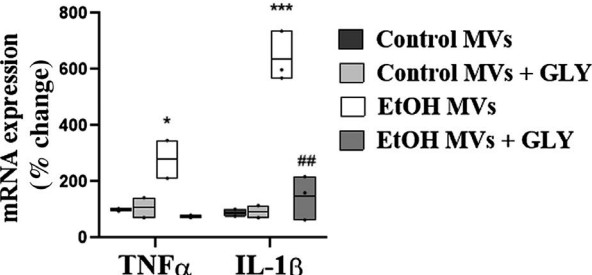

(a) **HMGB1 blockade prevents proinflammatory induction by MVs**

Legend:
- ■ Control MVs
- ▨ Control MVs + GLY
- □ EtOH MVs
- ▧ EtOH MVs + GLY

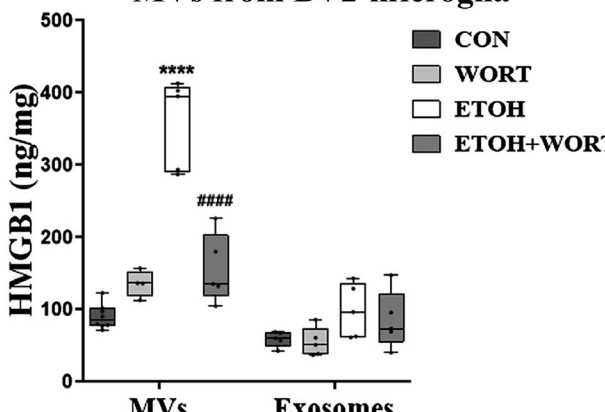

(b) **PI3K-mediated secretion of HMGB1 in MVs from BV2 microglia**

Legend:
- ■ CON
- ▨ WORT
- □ ETOH
- ▧ ETOH+WORT

**FIGURE 7** PI3K mediates the secretion of pro-inflammatory HMGB1-containing EtOH-MVs. (a) Primary organotypic brain slice cultures (OBSCs) were treated with ethanol and microvesicles (MVs) isolated by centrifugation. Pelleted MVs were incubated with either vehicle or the HMGB1 inhibitor glycyrrhizin (GLY, 200 µM for 30 min), then repelleted and resuspended in media. Conditioned vehicle or GLY-treated MVs were then added to naïve OBSC for 96 hr. EtOH-MVs induced TNF-α (twofold) and IL-1β (sixfold). Neutralization of HMGB1 in EtOH-MVs with GLY abolished their induction of TNF-α and IL-1β in naïve OBSC. (b) BV2 microglia were treated with ethanol ± the PI3K inhibitor wortmannin (WORT, 200 nM) or vehicle control for 24 hr. The MV fraction was isolated by final centrifugation at 21,000× g for 60 min and the exosomal fraction by centrifugation at 100,000× g × 2 hr. HMGB1 was measured by ELISA. Ethanol caused a fourfold increase in HMGB1 in the MV compartment with no significant change in the exosomal compartment. PI3K inhibition with WORT blocked the ethanol-induced increase in MV HMGB1. $N = 4–8$ wells/group, ****$p < 0.0001$ versus control, ####$p < 0.0001$ versus EtOH

activation (Dupont et al., 2011; Nickel & Rabouille, 2009; Vinod et al., 2014). Together, these data support a role for MV signaling in promoting pro-inflammatory responses to ethanol through PI3K-mediated release of DAMP-containing MVs.

Though it is now well known that ethanol induces pro-inflammatory signaling in brain, the underlying mechanisms that activate this signaling have been elusive. Reported evidence supports a

role for a host of factors such as metabolic disturbances, formation of reactive oxygen species (Qin & Crews, 2012), or effects on cellular membranes to promote TLR dimerization and subsequent activation (Fernandez-Lizarbe et al., 2013). However, ethanol also alters membrane composition (Saito et al., 2005) and fluidity (Goldstein, 1986), increasing ceramides which are key for membrane reorganization and MV secretion. aSMase catalyzes the production of ceramides from sphingomyelin during this process, and we find aSMase inhibition with IMP blocks pro-inflammatory gene induction by ethanol. It is important to note that IMP also blocks the reuptake of monoamines and blocks dopamine and acetylcholine receptors, which could contribute to its anti-inflammatory properties. For this reason, both inhibition of MV secretion and MV transfer experiments were employed according to ISEV guidelines to ascertain the role of MVs in ethanol-mediated induction of pro-inflammatory signaling. Though it is difficult to discern whether a single cellular disruption caused by ethanol is primarily responsible for the resultant pro-inflammatory activation, the combination of cellular insults seems to converge on the secretion of DAMP-containing MVs and subsequent induction of TLR signaling. TLR induction is a prominent feature both in human postmortem AUD brain tissue and in rodent models (Crews et al., 2017; Grantham et al., 2020; McCarthy et al., 2017; Montesinos et al., 2016). The nature of ethanol-induced pro-inflammatory cytokine induction downstream of TLR signaling (e.g., TNF-α and IL-1β) *in vivo* depends on the chronicity of alcohol treatment, the brain region, and the time point that is assessed. Our findings of an induction TNF-α and IL-1β in OBSC are consistent with previous reports using chronic ethanol models *in vivo* across multiple brain regions (Pascual et al., 2015; Patel et al., 2019; Qin et al., 2008) and with findings in postmortem human AUD hippocampus (Coleman, Zou, & Crews, 2017; Coleman, Zou, Qin, et al., 2017). However, in acute ethanol models *in vivo*, TNF-α and IL-1β are low at early time points (<10 hr) and are elevated at later time points into withdrawal (15–18 hr) (Doremus-Fitzwater et al., 2014; Walter & Crews, 2017). Thus, our current and previous studies in OBSC model findings in chronic ethanol treatments *in vivo* and in postmortem human AUD brain (Coleman, Zou, Qin, et al., 2017; Crews et al., 2013; Zou & Crews, 2014). The reason for this is unclear but could be due to the absence of factors from the peripheral circulation that contribute to differences in responses in acute versus chronic treatment *in vivo*. Nonetheless, this work supports that MVs promote or may even drive pro-inflammatory activation in brain by ethanol. In this study we used centrifugation to differentiate between exosomes and MVs; however, there is some overlap in size (50–150 nm) of these populations. We find HMGB1 secreted by microglia in response to ethanol is primarily in the 50–500 nm size fraction. This could include both exosomes and MVs, though the exosome-enriched (<50 nm) fraction had relatively little HMGB1. Also, we suspect the ethanol-induced reduction in MVs involves increased microglial engulfment of MVs, as microglial depletion prevented the ethanol-induced reduction in MVs. This will be examined closely in future studies.

Extracellular vesicles (both MVs and exosomes) have now emerged as key mediators of immune signaling both in the brain and the periphery across several diseases and biological processes (Buzas et al., 2014; Tetta et al., 2013). These subcellular mediators contain relatively high levels of nucleic acids, bioactive lipids, and proteins. In the brain, vesicles can also be visualized and isolated from the extracellular matrix, and are thought to contribute to pathology in Alzheimer's disease, ALS, and Parkinson's disease (Gallart-Palau et al., 2020; Perez-Gonzalez et al., 2012; Vassileff et al., 2020). Thus, MV signaling within the brain parenchyma is likely an important feature of neurobiological function. MV secretion could promote communication across brain regions, as brain-derived MVs can be released into and circulate in the CSF (Doeuvre et al., 2009) and are thought to promote the propagation of tau pathology across brain regions (Perez et al., 2019). Further, MVs in plasma may also represent a new class of biomarkers for Alzheimer's disease (Abner et al., 2016) or traumatic brain injury (Guedes et al., 2020). Since pharmacological inhibition of their secretion or uptake is a feasible approach (Catalano & O'Driscoll, 2020), MVs represent a potentially novel therapeutic target for neurological diseases. Further, EVs could be used as vehicles for drug delivery, or as direct treatment agents. Recently, exosomes derived from mesenchymal stem cells were found to reduce oxidative stress, neuroinflammation, and relapse-enhanced drinking in response to chronic ethanol when administered intranasally (Ezquer et al., 2019). Thus, exogenously administered vesicles can not only reach the brain but also exert key effector functions. These studies highlight that EVs are important mediators of immune dysfunction, biomarkers, and perhaps treatment vehicles for multiple disease states.

In summary, we find MVs are key mediators of pro-inflammatory signaling in brain tissue in response to ethanol and that microglia are necessary for generation of pro-inflammatory MVs. Future work will explore the ability of modulating MV signaling *in vivo* on neuroimmune function and alcohol-related behaviors.

## ETHICS STATEMENT

All protocols followed in this study were approved by the Institutional Animal Care and Use Committee of The University of North Carolina at Chapel Hill and were in accordance with the guide for the care and use of laboratory animals (Institute of Laboratory Animal Resources (Unites States). Committee on Care and Use of Laboratory Animals, n.d.).

## DECLARATION OF TRANSPARENCY

The authors, reviewers and editors affirm that in accordance to the policies set by the *Journal of Neuroscience Research*, this manuscript presents an accurate and transparent account of the study being reported and that all critical details describing the methods and results are present.

## CONFLICT OF INTEREST

The authors have no conflict of interest to declare.

## AUTHOR CONTRIBUTIONS

*Conceptualization*, F.T.C., J.Z., and L.G.C.; *Investigation*, J.Z. and L.G.C.; *Methodology*, J.Z. and L.G.C., *Writing – Original Draft*, J.Z. and L.G.C.; *Writing – Review & Editing*, F.T.C. and L.G.C.; *Funding Acquisition*, F.T.C. and L.G.C.; *Resources*, F.T.C.; *Validation*, J.Z. and L.G.C.; *Project Administration*, L.G.C., *Visualization*, L.G.C.; *Supervision*, F.T.C. and L.G.C.

## PEER REVIEW

The peer review history for this article is available at https://publo ns.com/publon/10.1002/jnr.24813.

## DATA AVAILABILITY STATEMENT

The data that support the findings of this study are available from the corresponding author upon reasonable request.

## ORCID

*Leon G. Coleman* 🆔 https://orcid.org/0000-0003-1693-3799

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

## SUPPORTING INFORMATION

Additional supporting information may be found online in the Supporting Information section.

Transparent Peer Review Report

Transparent Science Questionnaire for Authors

