## [Transparent Peer Review Report · Journal of Neuroscience Research]

Extracellular Microvesicles Promote Microglia-Mediated Proinflammatory Responses to Ethanol

Coleman, Leon; Crews, Fulton; Zou, Jian

Review timeline:

Submission date: 16 July 2020

Editorial Decision: Major Modification (30 September 2020)

Revision Received: 4 January 2021

Accepted: 1 February 2021

Editor 1: Alex S Marshall
Editor 2: Cristina Ghiani
Reviewer 1: Gregg Homanics
Reviewer 2: Terrence Deak

1st Editorial Decision

Decision letter
Coleman:

Dear Dr

Thank you for submitting your manuscript to the Journal of Neuroscience Research. We've now received the reviewer feedback and have appended those reviews below. As you will see, the reviewers find the question addressed to be of potential interest. Yet, they do not find the manuscript suitable for publication in its current form.

If you feel that you can adequately address the concerns of the reviewers, you may revise and resubmit your paper within 90 days. It will require further review. Please explain in your cover letter how you have changed the present version. If you require longer than 90 days to make the revisions, please contact Dr Cristina Ghiani (cghiani@mednet.ucla.edu). To submit your revised manuscript: Log in by clicking on the link below

(If the above link space is blank, it is because you submitted your original manuscript through our old submission site. Therefore, to return your revision, please go to our new submission site here (submission.wiley.com/jnr) and submit your revision as a new manuscript; answer yes to the question "Are you returning a revision for a manuscript originally submitted to our former submission site (ScholarOne Manuscripts)? If you indicate yes, please enter your original manuscript's Manuscript ID number in the space below" and including your original submission's Manuscript ID number (jnr-2020-Jul-8928) where indicated. This will help us to link your revision to your original submission.)

Thank you again for your submission to the Journal of Neuroscience Research; we look forward to reading your revised manuscript.

Best Wishes,

Dr Alex Marshall
Associate Editor, Journal of Neuroscience Research

Dr Cristina Ghiani
Editor-in-Chief, Journal of Neuroscience Research

Associate Editor: Marshall, Alex

Comments to the Author:

I apologize that it has taken so long to get your reviews in. I think that many reviewers have been overwhelmed recently. I hope that this delay has not impacted your productivity or interest in our journal.

-S. Alex Marshall

Additional Editorial Comments:

1. Please move the figure legends after the references.
2. JNR does not support the usage of bar graphs, kindly change all continuous data plots into scatterplots or box and whisker plots rather than bar graphs to better visualize the distribution of data.

3. CONFLICT OF INTEREST AND AUTHOR CONTRIBUTIONS

Please add to your paper (after the Discussion and Acknowledgments, immediately before the References) a conflict of interest statement and a statement of authors' contributions. The statement must follow the CRediT Taxonomy. You can find examples of such statements in the author guidelines on-line at [http://onlinelibrary.wiley.com/journal/10.1002/\(ISSN\)1097-4547/homepage/ForAuthors.html](http://onlinelibrary.wiley.com/journal/10.1002/(ISSN)1097-4547/homepage/ForAuthors.html)

4. GRAPHICAL ABSTRACT

Please upload a graphical abstract, which we are asking of all authors submitting original research articles. This is intended to provide readers with a visual representation of the conclusions and an additional way to access the contents and appreciate the main message of the work. What we require is a .tif image file and a .doc text file containing an abbreviated abstract. For the image, labels, although useful, must be kept to a minimum and the image should be 400 x 300, 300 x 400, or 400 x 400 pixels square and at a resolution of 72 dpi. This can be one of the figures from your article, or something slightly different, as long as it represents your study. Instructions for this can be found in our author guidelines online at [http://onlinelibrary.wiley.com/journal/10.1002/\(ISSN\)1097-4547/homepage/ForAuthors.html](http://onlinelibrary.wiley.com/journal/10.1002/(ISSN)1097-4547/homepage/ForAuthors.html)

Reviewer: 1

Comments to the Author

The authors describe their investigation into the role of extracellular vesicles (EVs) in neuroinflammatory signaling in response to ethanol exposure. Using organotypic brain slice culture (OBSCs), the authors were able to determine that ethanol exposure increases the proinflammatory cargo of EVs, and these EVs are sufficient to induce inflammatory pathways in naive OBSCs. Additionally, this process appears to be driven by microglial EVs, as elimination of these cells in culture blocks induction of the proinflammatory state. Overall, the findings of this article are very exciting for the alcohol field, as well as the neuroscience field as a whole. This being said, we have some suggestions that we believe would strengthen this report.

1. Authors should go over the manuscript closely for grammatical correctness and missing words. There are too many errors to list individually so we recommend the authors to review closely.
2. To make the manuscript more accessible for individuals not familiar with the proinflammatory pathway investigated, a more in-depth description should be included in the introduction outlining the key players and the enzymes/proteins that will be mentioned in the manuscript.
3. Throughout the manuscript, authors should replace RT-PCR with RT-qPCR.
4. p.5 line 40: please provide more specifics on type of "membrane tissue insert" used (e.g., Manufacturer, catalogue number).
5. p.5, line 53: the statement "containing 300ml water saturated with equal concentrations of ethanol" is confusing. Suggested rewording: 'containing 300ml water plus ethanol at the same concentration present in the culture media'.
6. p.6 line 25: "pellet from 2-3 culture media". Should this be 2-3 ml culture media, or something else?
7. p.6 line 25: capitalize Trizol
8. p.7: in the methods description of RT-qPCR, please include a section describing how you validated the primers for efficiency and specificity
9. p.7 line 6: capitalize Nanodrop
10. p.7 line 31 and p.5, line56: HEC slices are mentioned in the methods, but nowhere else in the paper. Please clarify.

11. Great confusion emerges from the inconsistent use of the terms EVs and MVs. For example, in the introduction (p.3, line 43) the authors define MVs as a subclass of EVs. However, elsewhere in the manuscript the terms appear to be used interchangeably (e.g., p.6, lines 20; lines 40-42). It is completely unclear if EVs or MVs were used for the experiments described in the manuscript. Please be clear and consistent.
12. p.8 line 21: 100 μ M should be 100mM. Same change needed in figure legends. The authors should check the entire manuscript for accuracy.
13. p.8, lines 22-24: 100mM ethanol is a very high concentration that is associated with coma or death in non-tolerant people. The authors should acknowledge this and better justify the use of such a high concentration ethanol (e.g., people with AUD have been reported to have 200-400mM BECs).
14. Statistical results are not presented consistently throughout the manuscript. Sometimes they appear in the results section, while other times they appear in the figure legends. Please be consistent throughout the manuscript.
15. p.9: For the RT-qPCR results described, please include what gene the results are normalized to (either β -action or 18S) either in the text or the figure legend.
16. Although imipramine blocks EV release, it also blocks reuptake of many neurotransmitters which could contribute to the observed effects. This limitation should be acknowledged in the manuscript and the conclusion offered on p.10, lines 19-23 should be modified.
17. p.10 line 37-39: Authors state "The EV isolation results in the removal of any remaining ethanol or residual media, meaning all observed effects are mediated solely by EVs." This is an overstatement of these findings as the authors cannot rule out an effect of contaminants such as protein aggregates that copurify with EVs.
18. p.10 line 45: Experiments using PKH67 to stain the EVs are presented, but no methods are provided. Please include a description of how PKH67 was incorporated into EVs and imaged in the methods section. Additionally, without a no EV + dye control, the authors cannot conclude that the EVs are required for dye transfer to brain. It is possible that residual unincorporated dye in the EV preparation was directly transferred to brain.
19. p.11 line 4: Very confusing. Please reword for clarity.
20. p.11 line 36-49: (a) Evaporation experiment was very confusing. Please rewrite for clarity. (b) A description of how EV depleted media was created was not included in the methods. (c) Additionally, an evaporated, EV intact control is required to rule out an effect of evaporation on some other component beside EVs that is responsible for the effect.
21. p.12 lines 13-17: Authors mention that a panel of 6 microglial genes were investigated to ensure that PLX3397 treatment removed microglia in OBSC culture and all 6 genes showed the expected reduction. However, gene expression in other cell types (e.g., neurons, astrocytes) must be analyzed to demonstrate that the drug selectively affected only microglia as the authors imply. It is possible the drug was toxic and/or suppressed gene expression in general.
22. This paper could potentially have greater impact if the authors broadened the discussion of EVs in pathological conditions. Alzheimer's is mentioned, but studies have also investigated EV changes following TBI and stroke, as well as the therapeutic benefit of EVs in treating symptoms including neuroinflammation, demyelination and oxidative stress.
23. Authors should make sure that all figures are consistent for their use of "EV" or "MV" in the experimental schematics and legends. For example, Fig. 1 consistently uses EV in the figure and legend. Fig. 5a states "isolate EVs and label", but the vesicle diagram shows MV, and the legend is EV.
24. In figure 2 legend, please clarify exactly what the control is. Is the control the zero timepoint or separate OBSC not treated with ethanol? If the control is the zero timepoint, how can one be sure that the changes observed are not simply an effect of time vs an effect of ethanol treatment? Controls should really be slices that were not treated with ethanol at each timepoint.

25. In figure 2 legend please reword how you present the statistics. ANOVAs do not give individual differences between groups, only main effects. Post-hoc analysis provides the individual differences. Rewrite the legend to reflect this.

26. Figure 3 legend: Trizol does not isolate mRNA, it isolates total RNA. Please rewrite to reflect this.

27. Figure 6: Panel C does not include a control NTA trace. Please include control trace with the PLX and PLX+EtOH traces for comparison.

Reviewer: 2

Comments to the Author

This manuscript utilized organotypic brain slice cultures to evaluate the impact of ethanol on microglia-mediated inflammation and the potential involvement of microvesicles as delivery systems for packaged RNA. The manuscript addresses an interesting question with some novel mechanisms being examined. Unfortunately, the manuscript is substantially lacking in elaboration of experimental details, to the point that it is very difficult to understand what was done and why it was done. Moreover, lack of details make it impossible to evaluate the scientific rigor of the studies because of the lack of transparency. Many examples of this are raised below.

1. Abstract refers to aSMase inhibitor, which has not been defined. This is also not the conventional drug action of imipramine, so it seems inappropriate to describe it this way.
2. First paragraph of introduction makes many scientific claims without use citations. This needs to be corrected. A more general problem with the writing is the failure to review and cite articles outside the authors own laboratory.
3. The last paragraph of the discussion reiterates the abstract in its entirety. This paragraph should be re-focused to describe the experimental design and hypotheses being tested. The summary of results should come later in the manuscript (not in the introduction).
4. The subject section is both incomplete and inaccurate. Test subjects were neonates (not pregnant dams) and no information on breeding practices, control of litter effects, housing conditions, etc are provided. Group sizes, number of slices per pup, and sex of the slices is not reported. This is entirely unacceptable and violates most fundamental aspects of scientific transparency and rigor.
5. The concentration of PLX drugs to deplete microglia are not reported in the methods.
6. NIH does not regulate care and use of animals. This is overseen by PHS.
7. Information on Materials is also inaccurate and incomplete. Company names are not defined (MCE, IBL). LPS is an agonist of TLR4, but this is not its only action. The serotype of LPS must be provided, as potency across serotypes is highly variable.
8. The section on Experimental design and rigor does not address either issue sufficiently. Supplying a diagram cannot substitute for a verbal description.
9. For the OBSC procedure, why was it necessary to incubate slices for 14 days in vitro? This seems highly unusual, with many of the advantages of slice culture being lost as cells continue to die.
10. Most studies employed only a single concentration of drugs, with no cited rationale.
11. A minor point: the real time RT-PCR used here was relative, not quantitative as reported on Page 7.
12. Virtually no details on BV2 microglia experiments were provided. Relying on other published methods does not satisfy requirements to report what was done in the present experiments.
13. It seems that many of the drug treatments were for multiple days in vitro. What was the rationale for exposing tissue slices to ethanol for 4 days? Does PLX3397 exposure for 7 days cause cell death in the tissue due to the absence of microglia?

14. The results section fails to report any statistical analyses that were conducted or the statistical parameters (F values, df, p, etc). While some of this is reported in the Figure captions, this is highly unconventional and not an acceptable practice for most journals.

15. Figure 2 shows induction of TNF and IL-1b after prolonged ethanol exposure in vitro. This is the opposite pattern that has been reported by many labs using in vivo studies. How do the authors explain the relevance of these findings when compared to data from their own lab (Walter & Crews, 2017) shows ethanol suppressing TNF?

16. All abbreviations should be defined at first use.

Overall, the manuscript is quite difficult to follow. I strongly recommend that a revised manuscript adhere to the ARRIVE guidelines for rigor and transparency to allow for a full evaluation of the experimental methods and outcomes.

Authors' Response

Additional Editorial Comments:

1. *Please move the figure legends after the references.*

Response: As requested, we have moved the figure legends to after the references.

2. *JNR does not support the usage of bar graphs, kindly change all continuous data plots into scatterplots or box and whisker plots rather than bar graphs to better visualize the distribution of data.*

Response: As requested, we changed continuous data plots to box and whisker plots.

3. *CONFLICT OF INTEREST AND AUTHOR CONTRIBUTIONS Please add to your paper (after the Discussion and Acknowledgments, immediately before the References) a conflict of interest statement and a statement of authors' contributions. The statement must follow the CRediT Taxonomy. You can find examples of such statements in the author guidelines on-line at [http://onlinelibrary.wiley.com/journal/10.1002/\(ISSN\)1097-4547/homepage/ForAuthors.html](http://onlinelibrary.wiley.com/journal/10.1002/(ISSN)1097-4547/homepage/ForAuthors.html)*

Response: As requested, we have added these statements

4. GRAPHICAL ABSTRACT

Please upload a graphical abstract, which we are asking of all authors submitting original research articles. This is intended to provide readers with a visual representation of the conclusions and an additional way to access the contents and appreciate the main message of the work. What we require is a .tif image file and a .doc text file containing an abbreviated abstract. For the image, labels, although useful, must be kept to a minimum and the image should be 400 x 300, 300 x 400, or 400 x 400 pixels square and at a resolution of 72 dpi. This can be one of the figures from your article, or something slightly different, as long as it represents your study. Instructions for this can be found in our author guidelines online at [http://onlinelibrary.wiley.com/journal/10.1002/\(ISSN\)1097-4547/homepage/ForAuthors.html](http://onlinelibrary.wiley.com/journal/10.1002/(ISSN)1097-4547/homepage/ForAuthors.html)

Response: As requested, we have uploaded the graphical abstract.

Reviewer: 1

The authors describe their investigation into the role of extracellular vesicles (EVs) in neuroinflammatory signaling in response to ethanol exposure. Using organotypic brain slice culture (OBSCs), the authors were able to determine that ethanol exposure increases the proinflammatory cargo of EVs, and these EVs are sufficient to induce inflammatory pathways in naïve OBSCs. Additionally, this process appears to be driven by microglial EVs, as elimination of

these cells in culture blocks induction of the proinflammatory state. Overall, the findings of this article are very exciting for the alcohol field, as well as the neuroscience field as a whole. This being said, we have some suggestions that we believe would strengthen this report.

Overall Response: We thank the review for these comments and for the suggestions that we feel have greatly improved our manuscript

1. Authors should go over the manuscript closely for grammatical correctness and missing words. There are too many errors to list individually so we recommend the authors to review closely.

Response: As recommended, we have reviewed the article for grammatical correctness. Corrections are in the track edits version of the manuscript.

2. To make the manuscript more accessible for individuals not familiar with the proinflammatory pathway investigated, a more in-depth description should be included in the introduction outlining the key players and the enzymes/proteins that will be mentioned in the manuscript.

Response: As recommended, we have added a paragraph to the introduction to provide a more in-depth description of the proinflammatory pathways investigated. Paragraph 2 of the introduction (page 4) now reads:

“The signaling pathways associated with ethanol-induced neuroinflammation involve complex cascades resulting in coordinated regulation of several genetic programs. Initiation of this signaling involves includes secretion of TLR-activating DAMPs such as HMGB1 and miRNA let-7b (Crews, Qin et al. 2013, Zou and Crews 2014, Coleman, Zou et al. 2017). TLR ligation leads to activation of proinflammatory gene regulating transcription factors such as nuclear factor kappa-light-chain-enhancer of activated B cells (NF- κ B) and interferon regulatory factor 7 (IRF7), and AP-1(Cui, Shurtleff et al. 2014, Crews, Lawrimore et al. 2017). These transcription factors lead to increased gene expression of several proinflammatory cytokines such as TNF α and IL-1 β . TLR activation also causes structural as well as functional changes in microglia, the predominant immune cell in the brain(Fernandez-Lizarbe, Pascual et al. 2009, Rosenberger, Derkow et al. 2014, Lawrimore and Crews 2017). At rest, microglia adopt a homeostatic/surveillance state that can change to hyper-ramified or amoeboid structural states that may feature a range of functional changes. Expression levels of microglial-enriched factors such as Tmem119, P2RY12, CX3CR1, C1q, C3R, and TREM2 (Butovsky, Jedrychowski et al. 2014) can change with different insults, exposures, or disease states and modulate different microglial functions. For instance, expression of Tmem119 and P2RY12 decline in mouse models of Alzheimer’s disease (AD) (Keren-Shaul, Spinrad et al. 2017), C1q is involved in microglial pruning of synapses and microglial activation of astrocytes (Stephan, Madison et al. 2013, Hong, Beja-Glasser et al. 2016), CX3CR1 regulates microglia activation state via interaction with neuron-derived fractalkine (CX3CL1) (Paolicelli, Bisht et al. 2014), and TREM2 is in induced in AD models and is involved in microglia phagocytosis (Keren-Shaul, Spinrad et al. 2017). Ethanol is known to cause temporal and dynamic changes in microglia structure and activation, though its effects on these key mediators is unknown.”

3. p.5 line 40: please provide more specifics on type of “membrane tissue insert” used (e.g., Manufacturer, catalogue number).

Response: As recommended, this line now reads (pg 7 line 10): “Slices (~25/pup) were transversely cut with McIlwain tissue chopper at a thickness of 375 μ m and placed onto a 30 mm diameter Millicell Low height culture insert (Millipore, PICMORG50), 10-13 slices/tissue insert.”

4. p.5, line 53: the statement “containing 300ml water saturated with equal concentrations of ethanol” is confusing. Suggested rewording: ‘containing 300ml water plus ethanol at the same concentration present in the culture media’.

Response: As recommended, we have made this correction.

5. p.6 line 25: “pellet from 2-3 culture media”. Should this be 2-3 ml culture media, or something else?

p.6 line 25: capitalize Trizol

Response: As recommended, this line (pg 9 line 11) now reads: “Briefly, MVs from 2-3 mL culture media were dissolved in total of 700ul Trizol.”

6. p.7: in the methods description of RT-qPCR, please include a section describing how you validated the primers for efficiency and specificity

Response: As recommended, we added the following statement to the methods (pg 9 line 22): “Primer sequences were validated using the NCBI Primer-BLAST, with only primers that targeted the desired mRNA and had single-peak melt curves (showing amplification of a single product) being used.”

7. p.7 line 6: capitalize Nanodrop

Response: As recommended, we have made this correction.

8. p.7 line 31 and p.5, line56: HEC slices are mentioned in the methods, but nowhere else in the paper. Please clarify.

Response: As recommended, we have corrected this abbreviation to read OBSC throughout the manuscript.

9. Great confusion emerges from the inconsistent use of the terms EVs and MVs. For example, in the introduction (p.3, line 43) the authors define MVs as a subclass of EVs. However, elsewhere in the manuscript the terms appear to be used interchangeably (e.g., p.6, lines 20; lines 40-42). It is completely unclear if EVs or MVs were used for the experiments described in the manuscript. Please be clear and consistent.

Response: We apologize for this confusion. MVs were isolated, assessed and used for transfer experiments. We have replaced the abbreviation “EVs” with “MVs” except in cases where all EVs are being discussed (i.e. MVs and exosomes).

10. p.8 line 21: 100 μ M should be 100mM. Same change needed in figure legends. The authors should check the entire manuscript for accuracy.

p.8, lines 22-24: 100mM ethanol is a very high concentration that is associated with coma or death in non-tolerant people. The authors should acknowledge this and better justify the use of such a high concentration ethanol (e.g., people with AUD have been reported to have 200-400mM BECs).

Response: Thank you. We have corrected this typo throughout the manuscript, and this line (pg 7 line 24) now reads: “For regular slice cultures, OBSC slices at 14DIV were exposed to ethanol (100mM) for different time points. Though this is a high concentration alcohol for non-dependent individuals, patients with AUD reach these blood alcohol concentrations while being conscious and functional (Olson, Smith et al. 2013), and we have previously found this concentration ex-vivo causes induction of immune genes that is similar to findings in vivo and in postmortem human alcohol use disorder brain (Crews, Qin et al. 2013, Coleman, Zou et al. 2017).”

11. Statistical results are not presented consistently throughout the manuscript. Sometimes they appear in the results section, while other times they appear in the figure legends. Please be consistent throughout the manuscript.

Response: As recommended, we now present statistical results in both the results and figure legends.

12. p.9: For the RT-qPCR results described, please include what gene the results are normalized to (either β -actin or 18S) either in the text or the figure legend.

Response: As recommended, we now state in the methods (pg 10 line 3): “For all analyses in OBSC tissue β -actin was used as the housekeeping gene. For measurement of mRNAs in MVs, 18S was used as the housekeeping gene since β -actin mRNA was not detected in the vesicles.” We also state the housekeeping gene in the Figure legends.

13. Although imipramine blocks EV release, it also blocks reuptake of many neurotransmitters which could contribute to the observed effects. This limitation should be acknowledged in the manuscript and the conclusion offered on p. 10, lines 19-23 should be modified.

Response: As recommended, we acknowledge this limitation and modified our conclusions. Page 13 line 24 in the Results now reads: “Thus, IMP blocked MV release and blunted proinflammatory gene induction by ethanol, consistent with MVs promoting ethanol-induced proinflammatory activation in brain. However, IMP also has effects on several neurotransmitter systems which could its anti-inflammatory effects. Therefore, we next employed ethanol-

conditioned MV transfer experiments to further clarify their role in ethanol-induction of proinflammatory responses.”

Also, page 19 line 2 in the Discussion now reads: “It is important to note that IMP also blocks reuptake of monoamines and blocks dopamine and acetylcholine receptors which could contribute to its anti-inflammatory properties. For this reason, both inhibition of MV secretion and MV transfer experiments were employed according to ISEV guidelines to ascertain the role of MVs in ethanol-induction of proinflammatory signaling.”

14. p.10 line 37-39: Authors state “The EV isolation results in the removal of any remaining ethanol or residual media, meaning all observed effects are mediated solely by EVs.” This is an overstatement of these findings as the authors cannot rule out an effect of contaminants such as protein aggregates that copurify with EVs.

Response: We appreciate with this critique and have reworded this sentence (pg 14 line 8) to more clearly state: “The MV isolation results in the removal residual media, meaning the observed effects are mediated by MVs or other pelleted proteins or lipids rather than any remaining ethanol.”

15. p.10 line 45: Experiments using PKH67 to stain the EVs are presented, but no methods are provided. Please include a description of how PKH67 was incorporated into EVs and imaged in the methods section. Additionally, without a no EV + dye control, the authors cannot conclude that the EVs are required for dye transfer to brain. It is possible that residual unincorporated dye in the EV preparation was directly transferred to brain.

Response: To address these concerns we have added details to the methods which on page 8 line 15 of the methods now reads: “For fluorescent labeling of MVs, MV pellets were labeled using PKH67 dye (Sigma-Aldrich, Cat# MIDI67-1KT), a fluorescent dye that labels the lipid membranes, according to the manufacturer’s instructions with some modifications. Briefly, the MV pellet was resuspended in 200µL of diluent C (provided by the manufacturer) and 6 µL of PKH67 was added. After incubation for 5 min at room temperature, labeling was stopped by addition of an equal volume of medium (200µL) containing 25% horse serum for 1 min. The labeled MVs were then re-pelleted and washed once with PBS to remove any residual dye. The final labeled MV pellet was resuspended with MEM containing 6% HS and applied to new slice cultures. Cellular uptake of labeled MVs was observed using a Leica Stellaris 5 confocal microscope.”

16. p.11 line 4: Very confusing. Please reword for clarity.

Response: As recommended this sentence (page 14 line 20) now reads: “We therefore compared the effects of control-MVs and EtOH-MVs on OBSCs treated for the same duration on: proinflammatory genes, microglial activation genes, and microglial functional genes.”

17. p.11 line 36-49: (a) Evaporation experiment was very confusing. Please rewrite for clarity. (b) A description of how EV depleted media was created was not included in the methods. (c) Additionally, an evaporated, EV intact control is required to rule out an effect of evaporation on some other component beside EVs that is responsible for the effect.

Response: We apologize for the confusion. As recommended, we have added text to the methods and clarified that both control and ethanol-treated groups underwent the same

evaporation and MV depletion by centrifugation. MVs were still present in the media during ethanol evaporation. There was no effect of the MV-depleted media on TNF α or IL-1 β gene induction in naïve brain slices. Page 8 line 26 in the methods now read: “*To ascertain the role of non-MV components, media was collected from control and ethanol-treated OBSC and stored in a sterile manner in the incubator overnight to allow remaining ethanol to evaporate from the media as we have previously reported (Walter & Crews, 2017). MVs were removed from either control or ethanol-conditioned media by centrifugation. MV-depleted media, in a 1:1 dilution with fresh media to ensure appropriate cellular nutrition and media pH, was added to naïve OBSC and proinflammatory gene expression measured.*”

18. p.12 lines 13-17: Authors mention that a panel of 6 microglial genes were investigated to ensure that PLX3397 treatment removed microglia in OBSC culture and all 6 genes showed the expected reduction. However, gene expression in other cell types (e.g., neurons, astrocytes) must be analyzed to demonstrate that the drug selectively affected only microglia as the authors imply. It is possible the drug was toxic and/or suppressed gene expression in general.

Response: We thank the Reviewer for this comment. Both we and others have previously performed these analyses in prior reports and found that PLX3397 has no measureable toxic effect on other cell types either *in vivo* or in the OBSC used here. Page 15 line 25 now reads, “*Microglia can be depleted both in vivo and ex vivo with pharmacological inhibition of CSF1R, with no detectable non-microglial cell death or behavioral dysfunction (Coleman et al., 2020; Han, Harris, & Zhang, 2017; Hughes & Bergles, 2014; Rice et al., 2015; Varvel et al., 2012; Walter & Crews, 2017).*”

19. This paper could potentially have greater impact if the authors broadened the discussion of EVs in pathological conditions. Alzheimer’s is mentioned, but studies have also investigated EV changes following TBI and stroke, as well as the therapeutic benefit of EVs in treating symptoms including neuroinflammation, demyelination and oxidative stress.

Response: We thank the Reviewer for this suggestion. As recommended, we have expanded the discussion and added a paragraph to discuss the role of vesicles in other diseases as well as potential therapeutic use. Paragraph 3 of the discussion now reads:

“*Extracellular vesicles (both MVs and exosomes) have now emerged as key mediators of immune signaling both in the brain and the periphery across several diseases and biological processes (Buzas et al., 2014; Tetta, Ghigo, Silengo, Deregibus, & Camussi, 2013). These subcellular mediators contain relatively high levels of nucleic acids, bioactive lipids, and proteins. In the brain, vesicles can be also visualized and isolated from the extracellular matrix, and are thought to contribute to pathology in Alzheimer’s disease, ALS, and Parkinson’s diseases (Gallart-Palau, Guo, Serra, & Sze, 2020; Perez-Gonzalez, Gauthier, Kumar, & Levy, 2012; Vassileff, Cheng, & Hill, 2020). Thus, MV signaling within the brain parenchyma is likely an important feature of neurobiological function. MVs secretion could promote communication across brain regions, as brain-derived MVs can be released into and circulate in the CSF (Doeuvre, Plawinski, Toti, & Angles-Cano, 2009) and are thought to promote propagation of tau pathology across brain regions (Perez, Avila, & Hernandez, 2019). Further, MVs in plasma may also represent a new*

class of biomarkers for Alzheimer's disease (Abner, Jicha, Shaw, Trojanowski, & Goetzl, 2016) or traumatic brain injury (Guedes et al., 2020). Since pharmacological inhibition of their secretion or uptake is a feasible approach (Catalano & O'Driscoll, 2020), MVs represent a potentially novel therapeutic target for neurological diseases. Further, extracellular vesicles could be used as vehicles for drug delivery, or as direct treatment agents. Recently, exosomes derived from mesenchymal stem cells were found to reduce oxidative stress, neuroinflammation, and relapse-enhanced drinking in response to chronic ethanol when administered intranasally (Ezquer et al., 2019). Thus, exogenously administered vesicles can not only reach the brain but also exert key effector functions. These studies highlight that extracellular vesicles are important mediators of immune dysfunction, biomarkers, and perhaps treatment vehicles that across multiple disease states.”

20. Authors should make sure that all figures are consistent for their use of “EV” or “MV” in the experimental schematics and legends. For example, Fig. 1 consistently uses EV in the figure and legend. Fig. 5a states “isolate EVs and label”, but the vesicle diagram shows MV, and the legend is EV.

Response: As requested we have made these corrections throughout the manuscript.

21. In figure 2 legend, please clarify exactly what the control is. Is the control the zero timepoint or separate OBSC not treated with ethanol? If the control is the zero timepoint, how can one be sure that the changes observed are not simply an effect of time vs an effect of ethanol treatment? Controls should really be slices that were not treated with ethanol at each timepoint.

Response: We thank the Reviewer for this observation. All treated time points were compared to control treatments at the same time point. For ethanol vs control treatments (Figure 2), there is no time effect on control untreated slices on innate immune gene activation over the 96h period, and for much longer periods as we recently reported (Coleman, Zou et al. 2020). Therefore, control slices at 96h were used. For MV transfer experiments, the zero-time point was in fact the zero time point prior to treatment with either control-MVs or ethanol-MVs. However, as recommended we have removed the zero-time point to avoid any confusion for the reader. Ethanol-MV values are presented as % of control-MV-induced changes at the same time point. To make this clearer we have added text to the results sections below.

Page 14 line 18 now reads: “Since MV transfer increases the total number of media MVs from baseline levels, all findings from EtOH-MVs were compared to MVs from DIV-matched control OBSC (control-MVs) to account for any potential non-specific effects caused by an increase in total media MVs. We therefore compared the effects of transferred control-MVs and EtOH-MVs treated for the same duration on: OBSC proinflammatory genes, microglial activation genes, and microglial functional genes.”

22. In figure 2 legend please reword how you present the statistics. ANOVAs do not give individual differences between groups, only main effects. Post-hoc analysis provides the individual differences. Rewrite the legend to reflect this.

Response: As recommended we have reworded the Figure Legends to reflect this. For example, the legend for Figure 2 now reads: A main effect of ethanol treatment on TNF α (ANOVA $F_{4,25}=10.44$, $p<0.0001$) and IL-1 β ($F_{4,25}=2.981$, $p=0.039$) gene expression was observed with post-hoc analysis finding TNF α induction reached significance by 2h and IL-1 β reached a significant 2-fold induction by 24h.

23. Figure 3 legend: Trizol does not isolate mRNA, it isolates total RNA. Please rewrite to reflect this.

Response: As recommended this now reads: OBSC were treated with ethanol (100 μ M) for 96h. MVs were isolated from media by centrifugation, total RNA isolated, and gene expression measured by RT-PCR.

24. Figure 6: Panel C does not include a control NTA trace. Please include control trace with the PLX and PLX+EtOH traces for comparison.

Response: As recommended we have included this control trace in Figure 6 as well as the total MV numbers for the control slice in the insert.

Reviewer: 2

Comments to the Author

This manuscript utilized organotypic brain slice cultures to evaluate the impact of ethanol on microglia-mediated inflammation and the potential involvement of microvesicles as delivery systems for packaged RNA. The manuscript addresses an interesting question with some novel mechanisms being examined. Unfortunately, the manuscript is substantially lacking in elaboration of experimental details, to the point that it is very difficult to understand what was done and why it was done. Moreover, lack of details make it impossible to evaluate the scientific rigor of the studies because of the lack of transparency. Many examples of this are raised below.

Overall Response: We thank the Reviewer for their rigorous review and their comments. We appreciate the complexity of the experiments and have significant efforts to clarify the experimental details to ensure others can easily replicate our experimental designs. We feel the revisions have greatly improved the manuscript. Detailed changes are listed below:

1. Abstract refers to aSMase inhibitor, which has not been defined. This is also not the conventional drug action of imipramine, so it seems inappropriate to describe it this way.

Response: As recommended this line in the abstract now reads: "MV signaling was implicated in this response as reduction of MV secretion by imipramine blocked proinflammatory TNF α and IL-1 β induction by ethanol, and ethanol-conditioned MVs reproduced the ethanol-associated immune gene signature in naive OBSC slices."

2. First paragraph of introduction makes many scientific claims without use citations. This needs to be corrected. A more general problem with the writing is the failure to review and cite articles outside the authors own laboratory.

Response: As recommended we have reworked the introduction to include appropriate references from several labs in the field.

3. The last paragraph of the discussion reiterates the abstract in its entirety. This paragraph should be re-focused to describe the experimental design and hypotheses being tested. The summary of results should come later in the manuscript (not in the introduction).

Response: As recommended, we now focus the last paragraph in the introduction on the experimental design and hypothesis and it now reads: “We previously reported that MVs house DAMPs such as HMGB1 and let-7b which are TLR agonists that could activate glia (Coleman Jr, Maile, Jones, Cairns, & Crews, 2018; Coleman, Zou, & Crews, 2017). Therefore, we hypothesized that MVs are fundamental mediators of proinflammatory activation in response to ethanol. Further, since microglia are key initial immune responders to ethanol, and microglial depletion blunts many immune responses associated with ethanol, we hypothesized that microglial depletion would block the secretion of proinflammatory MVs. Secretion of DAMPs such as HMGB1 in MVs utilizes an unconventional secretory mechanism that is dependent on PI3-kinase (PI3K)/Akt signaling (Dupont et al., 2011; Jiang, Dupont, Castillo, & Deretic, 2013; Nickel & Rabouille, 2009). Thus, we hypothesized that this pathway would be key for ethanol-induced secretion of proinflammatory MVs from microglia. In order to determine the role of MVs in proinflammatory activation by ethanol, we applied the step-wise approach recommended by the International Society on Extracellular Vesicles (ISEV, Figure 1) (Lotvall et al., 2014; They et al., 2018; Witwer et al., 2013). This involves analysis of MV contents, inhibition of MV secretion, and transfer of conditioned MVs. We utilized an organotypic brain slice culture (OBSC) model that contains all cell types in situ, undergoes functional synaptic maturation (Stoppini, Buchs, & Muller, 1991), and recapitulates inflammatory responses to ethanol found in vivo and in human postmortem brain (Coleman, Zou, & Crews, 2017; Coleman et al., 2020; J. Y. Zou & Crews, 2014). Our findings indicate that MVs play an important role in regulating proinflammatory responses to ethanol.”

4. The subject section is both incomplete and inaccurate. Test subjects were neonates (not pregnant dams) and no information on breeding practices, control of litter effects, housing conditions, etc are provided. Group sizes, and sex of the slices is not reported.

Response: As recommended page 7 line 3 now reads: “Primary organotypic brain slice cultures (OBSC) were prepared from the hippocampal-entorhinal cortex formation of postnatal day 7 pups using previously established techniques of Stoppini and colleagues (Stoppini et al., 1991) with modifications as we have described previously (J. Zou & Crews, 2006; J. Y. Zou & Crews, 2005). Briefly, pregnant Sprague Dawley rat mothers were obtained from Charles River (Raleigh, NC, USA). Pregnant mothers were single housed. Neonates (an average of 6 per litter) at postnatal day 7 were decapitated, the brain was removed, and hippocampal-entorhinal complex dissected in Gey’s buffer (Sigma-Aldrich, St. Louis, MO). Slices (~25/pup) were transversely cut with McIlwain tissue chopper at a thickness of 375 μ m and placed onto a 30 mm diameter Millicell low height culture insert (Millipore, PICMORG50), 10-13 slices/tissue insert. Slices from both sexes were pooled together. Slices were cultured with MEM containing 25 mM HEPES and Hank’s salts, supplemented with 25% horse serum (HS) + 5.5 g/L glucose + 2 mM L-glutamine in a humidified 5% CO₂ incubator at 36.5oC for 7 days in vitro (DIV), followed by 4DIV in MEM + 12% HS, and then slices were cultured with MEM + 6% HS. Two replicates were performed for each experimental condition, with each experiment repeated at least twice.”

5. The concentration of PLX drugs to deplete microglia are not reported in the methods

Response: As recommended, Page 10 line 8 states: “For microglia depletion experiments, OBSC slices at 4DIV were treated with CSF1R inhibitor PLX3397 (1 μ M) for 7 days in regular culture medium (MEM containing 25% HS), followed by 3DIV in MEM + 12% HS to deplete microglia. We used 1 μ M PLX3397 since we have previously reported that this successfully depletes >90% of microglia in our ex vivo slice culture model (Coleman et al., 2020)”

6. NIH does not regulate care and use of animals. This is overseen by PHS.

Response: As recommended Page 6 line 9 now reads: “All protocols followed in this study were approved by the Institutional Animal Care Use Committee of The University of North Carolina at Chapel Hill and were in accordance with PHS policies for the care and use of animals in research”

7. Information on Materials is also inaccurate and incomplete. Company names are not defined (MCE, IBL).

Response: As recommended page 6 line 13 now reads: “The colony stimulating factor 1 receptor (CSF1R) inhibitor PLX3397 was obtained from MedChemExpress (Monmouth Junction, USA). Imipramine (IMP) was purchased from Sigma. Wortmannin (WORT) was obtained from Calbiochem (USA). The HMGB1 ELISA was obtained from Immuno-Biological Laboratories Co. International (Hamburg, Germany).”

8. The section on Experimental design and rigor does not address either issue sufficiently. Supplying a diagram cannot substitute for a verbal description.

Response: As recommended, page 6 line 17 in the experimental design and rigor now reads: “This was an exploratory study with the overall experimental design as depicted in Figure 1. Briefly, OBSC were (A) treated with ethanol, and expression of microglial and proinflammatory genes was measured as well as the number and size distribution of media MVs. This was done with and without IMP, an inhibitor of MV secretion. In (B), experiments testing the effect of ethanol-induced MVs on proinflammatory and microglial genes are depicted. MVs were isolated from control or ethanol-treated slices and transferred to naïve OBSC. This was done with or without the HMGB1 inhibitor glycyrrhizin (GLY). Each ethanol or ethanol-MV treated group was compared to a control or control-MV exposed specimens treated for the same amount of time. All findings were repeated at least once to ensure reproducibility. All approaches for determining the role of MVs in ethanol neuroimmune induction were in accordance with guidelines for studying EV biology recommended by the ISEV (Lotvall et al., 2014; They et al., 2018; Witwer et al., 2013).”

9. For the OBSC procedure, why was it necessary to incubate slices for 14 days in vitro? This seems highly unusual, with many of the advantages of slice culture being lost as cells continue to die.

Response: As recommended we have added text to the methods on page 7 line 18 which states: “OBSC slices were incubated for a total of 14 days prior to treatment as described by Stoppini et al during which OBSC stabilize as they continue to show functional maturation of synapses (Stoppini et al., 1991). OBSC slices are long-lived with no detectable cell death up to 42 days in culture in our previous report (Coleman et al., 2020).”

10. Most studies employed only a single concentration of drugs, with no cited rationale.

Response: As recommended we now provide the rationales for the concentration of each drug used:

Regarding ethanol, page 8 line 1 reads: “we have previously found this concentration *ex vivo* causes induction of immune genes that is similar to findings *in vivo* and in postmortem human AUD brain (Coleman, Zou, Qin, & Crews, 2017; Crews et al., 2013).”

Regarding imipramine, page 13 line 12 reads: “We tested IMP concentrations 1-10 μ M, previously reported to cause aSMase inhibition and to reduce MV secretion (Bianco et al., 2009; Catalano & O'Driscoll, 2020).” Further, we now show data from both 1 and 10 μ M IMP studies assessing the inhibition of proinflammatory gene induction with ethanol by IMP (Figure 4C-D).”

Page 10 line 10 reads: “We used 1 μ M PLX3397 since we have previously reported that this successfully depletes >90% of microglia in our ex vivo slice culture model (Coleman, Zou et al. 2020).”

Page 17 line 15 reads: “HMGB1 secretion in response to LPS from peripheral monocytes has been reported to involve PI3K-mediated mechanism thought to involve unconventional secretory autophagy (Oh, Youn et al. 2009, Dupont, Jiang et al. 2011). For this reason, we investigated whether PI3K inhibition with wortmannin (WORT) at similar concentration to Oh et al (200nM) would reduce ethanol-induced HMGB1 secretion. We found that 100nM caused a slight reduction in media HMGB1 (not shown), while WORT (200nM) completely blocked secretion of HMGB1 in MVs in response to ethanol, returning MV HMGB1 concentration to control levels (Fig. 7B).”

Page 9 line 6 reads: “For HMGB1 inhibition, MVs were resuspended in GLY (200 μ M), a concentration we previously reported is effective at blunting ethanol-mediated induction of proinflammatory genes (J. Y. Zou & Crews, 2014)”

11. A minor point: the real time RT-PCR used here was relative, not quantitative as reported on Page 7.

Response: As recommended, we have corrected this statement.

12. Virtually no details on BV2 microglia experiments were provided. Relying on other published methods does not satisfy requirements to report what was done in the present experiments.

Response: As recommended, we have added to the BV2 methods on page 10 line 14 which now states: “BV2 microglial cultures: BV2 microglia (from ICLC #ATL03001; Genoa, Italy) were maintained in culture as we and others have described previously in standard cell culture conditions (Coleman et al., 2017; Lawrimore, Coleman, Zou, & Crews, 2019). Briefly, 3×10^5 cells were plated per well in 6-well culture plates in DMEM media with 10% FBS, 1x glutaMAX and 1x penicillin/streptomycin antibiotic (Life Technologies). Cells adhered overnight and were treated with ethanol the following morning.”

13. It seems that many of the drug treatments were for multiple days in vitro. What was the rationale for exposing tissue slices to ethanol for 4 days? Does PLX3397 exposure for 7 days cause cell death in the tissue due to the absence of microglia?

Response: In Figure 1 we provide a time course of ethanol treatment from 1h to 4 days. We have previously reported 4 days of ethanol in OBSC produces similar immune gene induction seen in postmortem human alcohol use disorder brain tissue (Crews, Qin et al. 2013, Coleman, Zou et al. 2017). In Figure 4, a 24-hour treatment was used since induction of TNF α and IL-1 β at 24h reached its maximum at this time point. As recommended, page 13 line 10 now reads: “OBSC were treated for 24 hours since this was the earliest time point with maximal induction of TNF α and IL-1 β (Fig. 2A)”.

Response: PLX3397 does not cause any appreciable non-microglial cell death. As recommended, Page 15 line 25 now reads, “Microglia can be depleted both in vivo and ex vivo with pharmacological inhibition of CSF1R, with no detectable non-microglial cell death or behavioral dysfunction (Coleman et al., 2020; Han, Harris, & Zhang, 2017; Hughes & Bergles, 2014; Rice et al., 2015; Varvel et al., 2012; Walter & Crews, 2017). We have reported recently that seven days of treatment with the CSF1R antagonist PLX3397 (1 μ M) safely removes >90% of microglia from OBSC and blocks proinflammatory gene induction with ethanol (Coleman et al., 2020).”

14. The results section fails to report any statistical analyses that were conducted or the statistical parameters (F values, df, p, etc).

Response: As recommended we now include the statistical parameters in the results section as well as in the figure legends.

15. Figure 2 shows induction of TNF and IL-1 β after prolonged ethanol exposure in vitro. This is the opposite pattern that has been reported by many labs using in vivo studies. How do the authors explain the relevance of these findings when compared to data from their own lab (Walter & Crews, 2017) shows ethanol suppressing TNF?

Response: *In vivo* studies find that acute ethanol induces TNF α at early time points and induces it later time points such as 15-18h after ethanol (Walter & Crews, 2017)(Doremus-Fitzwater et al., 2014). Chronic ethanol models *in vivo*, however, find induction of TNF across multiple brain regions, consistent with our findings here in brain slice culture (Pascual, Balino, Aragon, & Guerri, 2015; Patel et al., 2019; Qin et al., 2008). To clarify this important distinction, as well as the nature of our findings in brain slice culture page 19 lines 9-24 now read:

“TLR induction is a prominent feature both in human postmortem AUD brain tissue and in rodent models (Crews, Walter, Coleman, & Vetreno, 2017; Grantham et al., 2020; McCarthy et al., 2017; Montesinos et al., 2016). The nature of ethanol-induced proinflammatory cytokine induction downstream of TLR signaling (e.g. TNF α and IL-1 β) in vivo depends on the chronicity of alcohol treatment, the brain region, and the time-point that is assessed. Our findings of an induction TNF α and IL-1 β in OBSC are consistent with previous reports using chronic ethanol models in vivo across multiple brain regions (Pascual, Balino, Aragon, & Guerri, 2015; Patel et al., 2019; Qin et al., 2008) and with findings in postmortem human AUD hippocampus (Coleman, Zou, & Crews, 2017; Coleman, Zou, Qin, et al., 2017). However, in acute ethanol models in vivo, TNF α and IL-1 β are low at early time points (<10h) and are elevated at later time points into withdrawal (15-18h) (Walter & Crews, 2017)(Doremus-Fitzwater et al., 2014). Thus, our current and previous studies in OBSC model findings in chronic ethanol treatments in vivo and in postmortem human AUD brain (Coleman, Zou, Qin, et al., 2017; Crews et al., 2013; J. Y. Zou & Crews, 2014). The reason for this is unclear but could be due to the absence of factors from the peripheral circulation that contribute to differences in responses in acute vs chronic treatment in vivo. Nonetheless, this work supports that MVs promote or may even drive proinflammatory activation in brain by ethanol.”

16. All abbreviations should be defined at first use.

Response: As recommended all abbreviations are defined at first use.

17. Overall, the manuscript is quite difficult to follow. I strongly recommend that a revised manuscript adhere to the ARRIVE guidelines for rigor and transparency to allow for a full evaluation of the experimental methods and outcomes.

Response: As recommended, we have significantly modified the text and added a paragraph to the end of the methods sections to show our compliance with the 2020 ARRIVE guidelines for rigor and transparency in animal studies that are relevant to the *ex-vivo* studies used here. This includes: study design, sample size, outcome measures, statistical methods, experimental animals, experimental procedures and results. This paragraph reads:

“Study Design, Sample Size, Statistical Methods, and Results: *For each experiment, 9-10 slices pooled slices were used per group with 2 technical replicates. The mean (\pm SEM) in each box plot is the average value across all experimental replicates. For each ethanol treatment experiment, ethanol-treated groups were compared to untreated control slices, with 2-8 experimental replicates. As we have reported previously, the genes measured did not change in untreated slices from 0h to the longest treatment duration (96h) (Coleman et al., 2020). Therefore,*

control slices at the 96h time point were used, and groups were analyzed by 1-way ANOVA with Dunnett's multiple comparisons test. The Brown-Forsythe test was used when the standard deviations were calculated to be significantly different across groups (GraphPad Prism). For MV-transfer experiments, slices treated with ethanol-conditioned MVs (i.e. MVs from ethanol-treated slice cultures), were compared to slices treated with control-conditioned MVs (i.e. MVs from control, untreated slices) for the same duration, with 3-8 experimental replicates and analyzed by 2-Way ANOVA with Sidak's multiple comparisons test. A p-value less than 0.05 was considered significant. This is consistent with the core set of standards for rigorous reporting issued by the NIH (Landis et al., 2012)."

2nd Editorial Decision

Decision Letter

Dear Dr Coleman:

Thank you for re-submitting your manuscript and the diligent efforts considering the original reviewer comments.

You will be pleased to know that your manuscript has been accepted for publication. Thank you for submitting this excellent work to our journal.

There are still some minor revisions suggested by one of the reviewers, it will not require a full resubmission, but please be sure to make the corrections in your proofs or if you like please send an updated version to the editorial office (jnroffice@wiley.com), they can upload it on your behalf.

In the coming weeks, the Production Department will contact you regarding a copyright transfer agreement and they will then send an electronic proof file of your article to you for your review and approval.

Please note that your article cannot be published until the publisher has received the appropriate signed license agreement. Within the next few days, the corresponding author will receive an email from Wiley's Author Services asking them to log in. There, they will be presented with the appropriate license for completion. Additional information can be found at <https://authorservices.wiley.com/author-resources/Journal-Authors/licensing-open-access/index.html>

Would you be interested in publishing your proven experimental method as a detailed step-by-step protocol? Current Protocols in Neuroscience welcomes proposals from prospective authors to disseminate their experimental methodology in the rapidly evolving field of neuroscience. Please submit your proposal here: <https://currentprotocols.onlinelibrary.wiley.com/hub/submitproposal>

Congratulations on your results, and thank you for choosing the Journal of Neuroscience Research for publishing your work. I hope you will consider us for the publication of your future manuscripts.

Sincerely,

Dr S. Alex Marshall
Associate Editor, Journal of Neuroscience Research

Dr Cristina Ghiani
Editor-in-Chief, Journal of Neuroscience Research

Associate Editor: Marshall, S. Alex
Comments to the Author:

Thanks so much for your diligent efforts considering the original reviewer comments. There are some minor revisions that should not require a full resubmission, but please be sure to make the corrections in your proofs.

Reviewer: 1

Comments to the Author

The revised manuscript is greatly improved. The authors have addressed all issues previously raised.

Reviewer: 2

Comments to the Author

The authors describe original work investigating microvesicles and neuroinflammatory signaling following ethanol exposure in organotypic brain slice cultures. They found that microvesicles promote proinflammatory activation following ethanol in a microglia-dependent manner. These novel findings have implications for not only alcohol use disorder but also other neurological diseases such as Alzheimer's disease. Importantly, the authors have made substantial changes to the initial version of the manuscript and responded to the concerns raised during previous reviews, all of which has greatly improved the manuscript as a whole. There are a few minor suggestions below which should be addressed prior to publication:

- Editing, particularly the references and newly added text

-Typos were found regarding the ethical approval paragraph in the method section and in the section on "Inhibition of MV secretion blocks ethanol-induced proinflammatory gene activation" towards the bottom of the paragraph "IMP blocked ethanol..."

-Care should also be taken to ensure the location for each supplier is listed the first time each supplier is named